# ORBIT: A Prognostic World Model
# for Ocular Reasoning Based on Imagined Trajectories

**Jiangtao Yan** [1 2] **Yanlin Qu** [3 4 5] **Yansheng Qiu** [1] **Shujian Gao** [4] **Wei Yu** [6 1] **Zheng Wang** [1 6 †] **Xiaodong Sun** [3 5]
**Huixun Jia** [3 5 †] **Diping Song** [2 †]

## Abstract

The longitudinal management of blinding fundus diseases constitutes a Partially Observable Markov Decision Process (POMDP) necessitating a critical precision-risk trade-off between intervention and over-treatment, as true pathology is often obscured in static observations. However, existing paradigms fail to address this complexity. Traditional vision models remain uninterpretable and memoryless, and while Vision-Language Models (VLMs) excel in semantic understanding, they rely on unsafe open-loop text reasoning lacking the anatomical grounding essential for clinical safety. Furthermore, robust learning is hindered by the scarcity of process supervision in sparse clinical records. To bridge this gap, we introduce the Logic-Constrained Abductive Data Engine. Operating on a "Propose-and-Verify" paradigm, it validates MLLM-proposed biomarkers against clinical and temporal logic to reconstruct dense pathological states from sparse outcomes. Building on this foundation, we propose ORBIT, the first ophthalmic Prognostic World Model. Uniquely, ORBIT employs counterfactual visual foresight to imagine anatomical futures under different treatments, anchoring decisions in Closed-Loop Anatomical Verification rather than linguistic probabilities. Experiments demonstrate that ORBIT effectively captures disease evolution and establishes a new paradigm for human-in-the-

loop longitudinal decision support and anatomically grounded treatment planning.

## 1. Introduction

Longitudinal management of blinding fundus diseases presents a critical clinical challenge (Cheung, 2025; Rajpurkar et al., 2022). Currently, anti-VEGF therapy stands as the cornerstone of treatment for blinding neovascular conditions. However, physicians operate under a precision-risk trade-off: balancing timely anti-VEGF intervention to avert irreversible vision loss against the cumulative risks of retinal atrophy and the socioeconomic burden of over-treatment (Nanji et al., 2025).

This task is fundamentally distinct from static diagnosis due to the intrinsic ambiguity of instantaneous observations. At a single time point, stable residual fluid and active recurrent leakage are often morphologically indistinguishable (Zur et al., 2025). Consequently, pathological activity acts as a latent variable that must be inferred from temporal dynamics rather than static features. Recognizing this unobservability, we formalize the problem as a Partially Observable Markov Decision Process (POMDP) (Hahsler & Cassandra, 2025), identifying the hidden state through a triplet of coupled **sub-tasks**: (1) **Perception** (extracting biomarkers), (2) **Evolution** (inferring dynamics), and (3) **Decision** (formulating treatment strategies).

There remains a substantial mismatch between pressing clinical demands and existing research paradigms, which cannot be directly adapted to this setting. On one hand, traditional vision models (Siméoni et al., 2025; Qiu et al., 2023a;b) ( Fig. 1A) function as memoryless black-boxes; they inherently lack the capacity to capture complex transition dynamics (LeCun et al., 2015). On the other hand, while VLMs (Lin et al., 2025; Kalpélbé et al., 2025) (Fig. 1B) excel in semantic understanding, they fundamentally rely on open-loop text reasoning. Without closed-loop anatomical verification, these models suffer from hallucination, leading to a critical misalignment between generated text and the underlying anatomical ground truth.

† Corresponding author. [1]National Engineering Research Center for Multimedia Software, School of Computer Science, Wuhan University, and School of Cyber Science and Engineering, Wuhan University [2]Shanghai Artificial Intelligence Laboratory [3]Shanghai General Hospital [4]Fudan University [5]Shanghai Jiaotong University [6]School of Artificial Intelligence, Wuhan University. Correspondence to: Zheng Wang <wangzwhu@whu.edu.cn>, Huixun Jia <jiahuixun@sjtu.edu.cn>, Diping Song <songdiping@pjlab.org.cn>.

*Proceedings of the 43$^{rd}$ International Conference on Machine Learning*, Seoul, South Korea. PMLR 306, 2026. Copyright 2026 by the author(s).

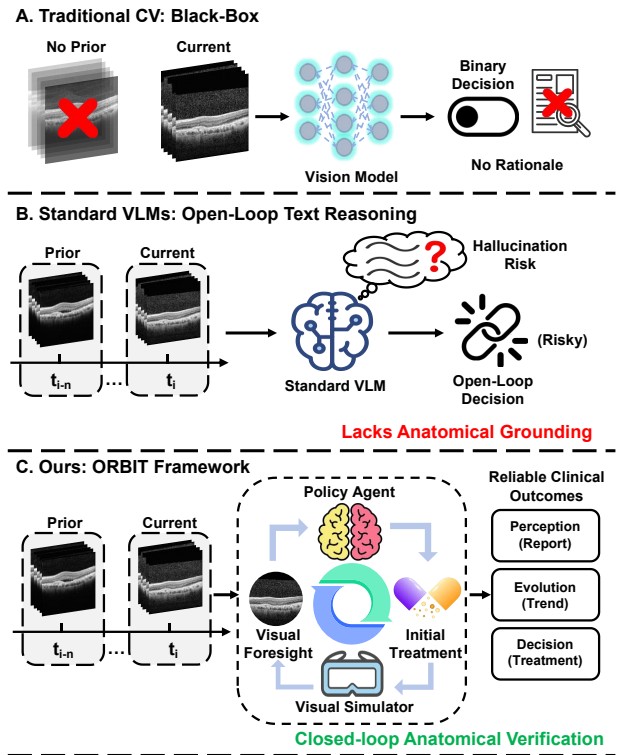

*Figure 1.* Comparison of clinical decision paradigms. **(A) Traditional Vision Models** operate as memoryless Black-Boxes, yielding binary decisions without rationale. **(B) Standard VLMs** incorporate temporal context but rely on open-loop text reasoning. **(C) ORBIT (Ours)** introduces a prognostic world model. It leverages visual foresight to simulate pathological evolution, enabling closed-loop anatomical verification for outcome-oriented planning.

Learning these complex dynamics, moreover, is hindered by a scarcity of process supervision (Yun et al., 2025). Clinical records typically contain sparse binary outcomes (e.g., injection decisions) but lack dense annotations of intermediate pathological states (Zhuang et al., 2025). To bridge this gap, we introduce the Logic-Constrained Abductive Data Engine. Adopting a "Propose-and-Verify" paradigm, it utilizes clinical outcomes and temporal consistency to rigorously validate LLM-proposed candidates, enabling the reconstruction of dense pathological states from sparse records.

Building upon this data-efficient foundation, we propose ORBIT (Fig. 1C), the first Prognostic World Model tailored for ophthalmology. Inspired by the human cognitive process of "mental simulation," ORBIT moves beyond passive state classification to perform active Counterfactual Visual Foresight. By performing forward rollouts (Dong et al., 2025; Gao et al., 2024) to synthesize future anatomical states (Phipps et al., 2025), ORBIT visually simulates pathological evolution conditioned on varying treatment policies. This capability advances ophthalmic AI from open-loop generation to robust planning grounded in Anatomical Verification.

Our main contributions are summarized as follows:

- **Problem Formulation:** We formulate ophthalmic disease management as a longitudinal decision-making problem under partial observability. This formulation enables the use of world-model-based counterfactual planning for sequential disease monitoring and treatment recommendation.

- **Data Strategy:** We introduce a **Logic-Constrained Abductive Data Engine** for deriving dense biomarker supervision from limited process-level annotations. The engine follows a "Propose-and-Verify" procedure and incorporates clinical-decision and temporal-consistency discriminators to improve label reliability.

- **Prognostic World-Model Architecture:** We propose **ORBIT**, a prognostic world model that combines visual foresight with anatomical verification. This design provides a structured framework for interpretable decision support in ophthalmic disease management, with the goal of improving reliability in complex longitudinal settings.

## 2. Related Work

### 2.1. Vision-Language Models in Medicine

Generalist Vision Language Models (VLMs) (Li et al., 2025b; Liu & Song, 2025; Team, 2025a; Gao et al., 2026; Qiu et al., 2025; Li et al., 2025a) have revolutionized static diagnostics by offering powerful zero-shot recognition and semantic report generation (Wang et al., 2025d). However, this "open-loop" text reasoning is ill-suited for longitudinal management because it fails to predict the causal impact of interventions (Kıcıman et al., 2024). Existing models lack counterfactual visual foresight and cannot simulate how anatomy evolves under different treatments, such as fluid absorption versus atrophy (Moor et al., 2023). Consequently, their decisions are driven by linguistic probabilities rather than patient-specific biological verification, rendering them unsafe for precise therapeutic planning (Van Veen et al., 2024).

### 2.2. World Models and Visual Foresight

World models (Ha & Schmidhuber, 2018; LeCun, 2022) enable counterfactual reasoning (Singh et al., 2025; Ma et al., 2025) by modeling state transitions $P(s_{t+1}|s_t, a_t)$. While MeWM (Yang et al., 2025) pioneered this for solid tumors, its restriction to irreversible morphological changes renders it inadequate for the stochastic fluid dynamics of retinal diseases. Unlike static ablation, retinal pathologies are governed by reversible and volatile pharmacodynamics (Tan et al., 2024). We propose ORBIT, the first Prognostic World

Model for ophthalmology, which extends the modeling horizon to long-term spatiotemporal evolution, enabling closed-loop visual foresight for chronic trajectories.

### 2.3. Reflective Reasoning in Medical Agents

Test-time compute strategies, such as Chain-of-Thought (CoT) (Wei et al., 2022) and Tree of Thoughts (ToT) (Yao et al., 2023), have empowered medical agents to decompose complex diagnostics into logical steps. Yet, this text-centric paradigm suffers from a critical limitation: a reliance on internal linguistic consistency. As inherently "open-loop" systems, they validate reasoning coherence rather than anatomical grounding, creating risks of circular reasoning (Dziri et al., 2023) where models hallucinate pathological features to rationalize incorrect decisions.

ORBIT overcomes this textual confinement via cross-modal anatomical verification. Diverging from purely semantic validation, we employ counterfactual visual foresight to ground textual hypotheses in simulated visual futures. This approach shifts the verification standard from semantic plausibility to anatomical fidelity, ensuring diagnostic reasoning is not only logically sound but also physically verifiable against the patient's specific pathological dynamics.

## 3. Data Construction

In clinical archives, therapeutic decisions are usually recorded, but the underlying biomarker evidence motivating these decisions is often absent, which creates a supervision gap and makes it difficult to interpret black-box predictions in terms of medical reasoning.

To bridge this gap, we propose a **Logic-Constrained Abductive Data Engine** shown in Fig. 2, and construct a new longitudinal ophthalmic decision dataset, **ORBIT-LD** (**ORBIT L**ongitudinal **D**ecision Dataset). Surpassing the limitations of traditional weak supervision, our approach reconstructs pathological states via a **"Propose-and-Verify"** paradigm. Instead of enforcing rigid outcome-to-feature rules, the engine first leverages Multimodal LLMs to propose candidate biomarkers based on visual evidence. These candidates are subsequently verified using clinical outcomes and temporal consistency as logical constraints. This mechanism effectively filters hallucinations, thereby improving the biological plausibility and self-consistency of the generated labels. The complete procedure, implementation details, and statistics are provided in Appendix A.

**Module 1: Hierarchical Semantic Synthesis (Propose).** We employ a coarse-to-fine Multimodal LLM pipeline to process volumetric OCT data. By aggregating local visual evidence from representative B-scans, the module synthesizes a candidate biomarker set $S_t$ and a preliminary structured report.

**Module 2: Dual-Consistency Verification Gates (Verify).** To ensure biological plausibility, synthesized reports are subjected to logical constraints. The Decision-Biomarker Consistency Gate enforces causal alignment between pathology and treatment. Specifically, interventional decisions unsupported by pathological findings are strictly rejected:

$$y_t = \text{INJECT} \ \wedge \ S_t = \emptyset \ \Rightarrow \ \text{INVALID}. \tag{1}$$

Conversely, cases of potential residual pathology ($y_t = \text{OBSERVE} \wedge S_t \neq \emptyset$) are routed for expert review. Subsequently, the Temporal Plausibility Gate regularizes longitudinal consistency by flagging abrupt biomarker substitutions that may indicate temporally inconsistent generation. Specifically, it identifies cases where at least one previously observed biomarker disappears while at least one new biomarker emerges between consecutive visits:

$$(S_{t-1} \setminus S_t \neq \emptyset) \wedge (S_t \setminus S_{t-1} \neq \emptyset). \tag{2}$$

This condition is used as a high-risk trigger rather than a hard rejection rule: flagged samples are routed to human-in-the-loop verification to distinguish genuine rapid pathological changes from generation errors. In addition, the gate restricts excessive mutations in the total biomarker burden:

$$\big|\, |S_t| - |S_{t-1}| \,\big| \geq 3. \tag{3}$$

Samples violating temporal constraints undergo human-in-the-loop verification to distinguish valid rapid progression or treatment response from generation errors.

**Module 3: Trajectory Modeling.** Finally, verified sequences are processed by a Pairwise Comparator to assign longitudinal evolution labels (Improved, Stable, Worsened), yielding the final ORBIT-LD dataset, with detailed dataset statistics and analysis provided in Appendix B.

**Gate Audit.** We audit 540 generated samples to quantify the effect of the verification gates. The Decision–Biomarker Consistency Gate rejects 29 severe decision–biomarker inconsistencies and routes 123 high-risk cases to expert review, while the Temporal Plausibility Gate flags 48 implausible longitudinal transitions. This audit suggests that the logic gates reduce high-risk errors, with expert review retained for ambiguous cases. Details are provided in Appendix A.6.

## 4. Methodology

### 4.1. Problem Formulation

We formalize longitudinal disease management as a Partially Observable Markov Decision Process (POMDP), defined by the tuple $\mathcal{M} = \langle \mathcal{S}, \mathcal{A}, \mathcal{O}, \mathcal{T}, \mathcal{R}, \gamma \rangle$. Here, $\mathcal{S}$ denotes the latent pathological ground truth (e.g., fluid activity), which is fundamentally obscured in high-dimensional static observations $\mathcal{O}$.

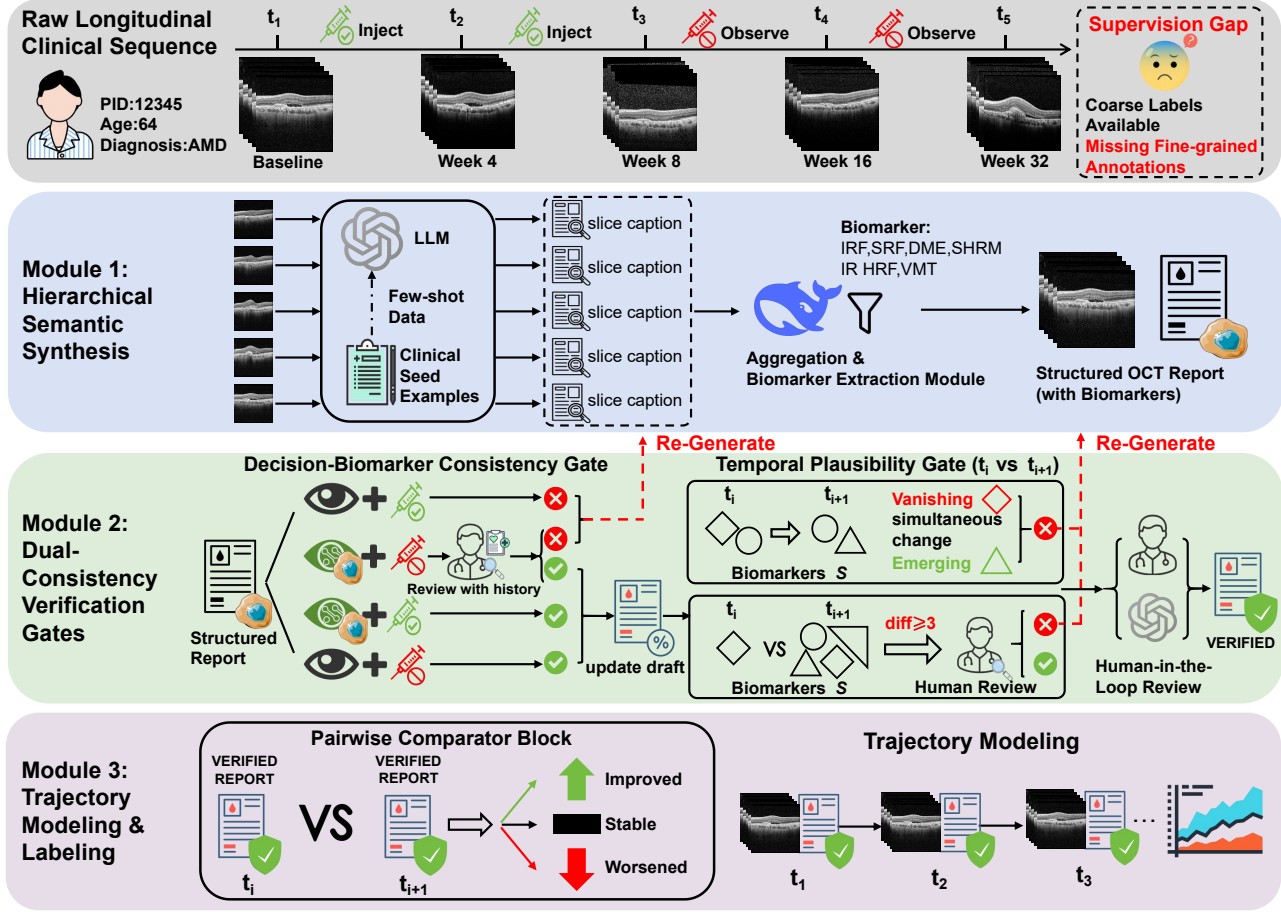

*Figure 2.* **The Logic-Constrained Abductive Data Engine.** The pipeline bridges **the clinical supervision gap** via three stages: **Module 1** synthesizes structured OCT reports via coarse-to-fine aggregation; **Module 2** ensures biological plausibility through the **Decision-Biomarker Consistency Gate** and the **Temporal Plausibility Gate** to filter hallucinations; and **Module 3** assigns evolution labels via a Pairwise Comparator for trajectory modeling.

Since the true pathological evolution $\mathcal{T}(s'|s, a)$ is inaccessible, we resolve this by learning a Prognostic World Model (Ha & Schmidhuber, 2018), parameterized by $\phi$, to act as a surrogate environment that approximates the stochastic dynamics:

$$\hat{\mathcal{T}}_\phi(\hat{s}_{t+1}|\hat{s}_t, a_t) \approx \mathcal{T}(s_{t+1}|s_t, a_t) \tag{4}$$

By internalizing these dynamics, the world model enables the policy $\pi_\theta(a|h)$ to maximize the expected cumulative return $J(\pi) = \mathbb{E}_{\tau \sim \pi} \left[\sum_{t=0}^{\infty} \gamma^t R(s_t, a_t)\right]$ via counterfactual reasoning, rather than reacting to instantaneous observations.

### 4.2. The ORBIT Architecture

We propose **ORBIT** as a concrete instantiation of the prognostic world model. As illustrated in Fig. 3A, the framework decouples dynamics representation and decision-making into two interacting modules:

**The Policy Agent** ($\pi_\theta$)**.** We instantiate the agent by fine-

tuning Qwen3-VL-8B-Instruct (Team, 2025b). Operating as a multimodal reasoner, it explicitly factorizes the decision process into five atomic steps (Madaan et al., 2023). As illustrated in Fig. 3A, the joint distribution is modeled as a causal chain:

$$\pi_\theta(\tau|o_t) = \underbrace{P(s_{\text{perc}}|o_t)}_{\text{Perception}} \cdot \underbrace{P(s_{\text{evol}}|s_{\text{perc}}, h_{t-1})}_{\text{Evolution}} \cdot \underbrace{P(a_{\text{hyp}}|s_{\text{evol}})}_{\text{Foresight}}$$
$$\cdot \underbrace{P(s_{\text{refl}}|a_{\text{hyp}}, \hat{I}_{\text{next}})}_{\text{Reflection}} \cdot \underbrace{P(a_{\text{final}}|s_{\text{refl}})}_{\text{Decision}} \tag{5}$$

This factorization enforces a strict reasoning loop. Specifically, the evolution step ($s_{\text{evol}}$) performs a comparative analysis between the current perception and history $h_{t-1}$ to classify the qualitative disease trend (i.e., *Improved*, *Stable*, or *Worsened*), which serves as the condition for subsequent decision-making.

**The Visual Simulator** ($f_\phi$)**.** To enable precise counterfactual reasoning, we train a diffusion-based Probabilistic

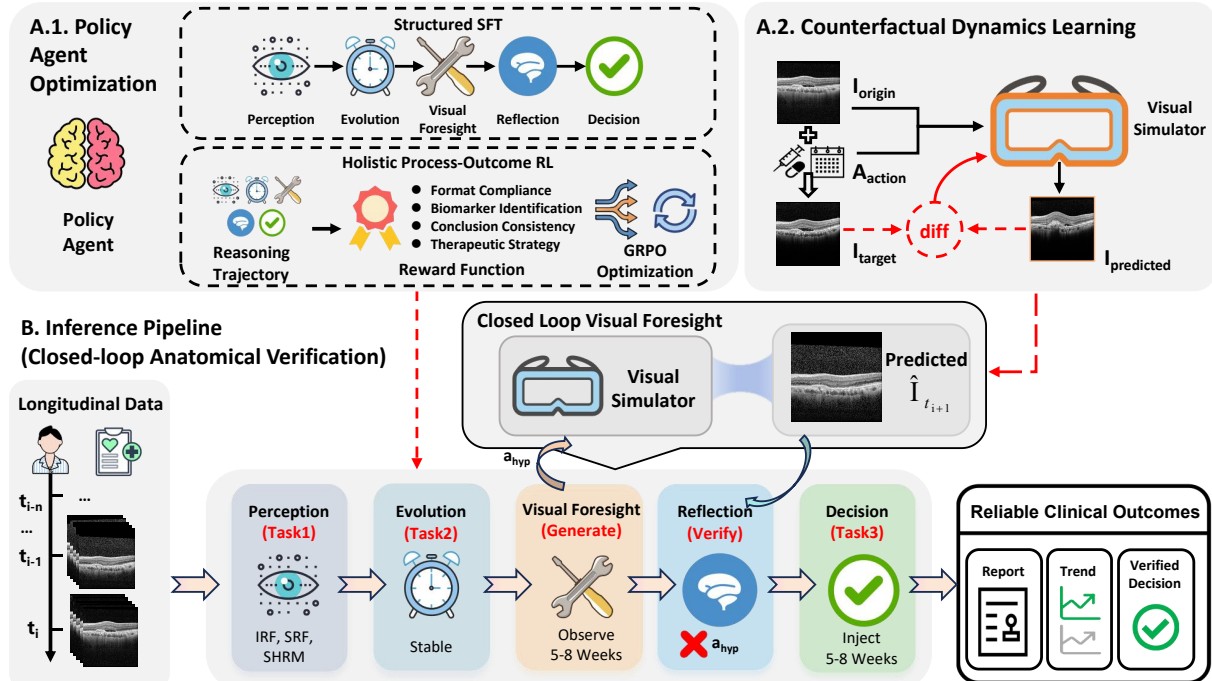

*Figure 3.* **The ORBIT Framework.** (A) **Training Paradigm**: The **Policy Agent** ($\pi_\theta$, instantiated via Qwen3-VL-8B) is optimized through a two-stage curriculum: **Structured SFT** to internalize the Chain-of-Thought reasoning protocols, and **Holistic Process-Outcome RL (GRPO)** to enforce multi-grained logical alignment. In parallel, the **Visual Simulator** ($f_\phi$) serves as a Probabilistic Transition Model, employing instruction-guided diffusion to learn pharmacodynamic responses. (B) **Inference Pipeline**: ORBIT executes the **Hypothesize-Simulate-Rectify Pipeline**. The agent first **Hypothesizes** a treatment plan ($a_{\text{hyp}}$) based on the disease trend (**Evolution**), then triggers the simulator to **Simulate** a counterfactual prognosis ($\hat{I}_{\text{next}}$) via visual rollout. Finally, a **Safety Critic** module **Rectifies** the decision if high-risk anatomical features are detected, enabling robust model-based verification.

Transition Model on top of the Qwen-Image-Edit architecture (Wu et al., 2025). It serves as a learnable environment proxy, predicting the future anatomical state $\hat{I}_{t+1}$ conditioned on the current observation $O_t$ and a proposed treatment action $a_{hyp}$.

### 4.3. Policy Agent Optimization

To synergize expert behavioral patterns with outcome-oriented logic, we optimize the Policy Agent via a two-stage training curriculum (detailed implementation configurations are provided in Appendix C):

**Stage 1: Structured SFT.** We first align the agent with expert protocols via Supervised Fine-Tuning (SFT). Leveraging the dense annotations in ORBIT-LD, we synthesize Chain-of-Thought (CoT) trajectories to explicitly model the reasoning process (Wang et al., 2025a; Qiu et al., 2026). This ensures the agent internalizes the clinical causal chain.

**Stage 2: Holistic Process–Outcome Reinforcement Learning.** Starting from the SFT-initialized policy, we further optimize the agent with Group Relative Policy Optimization (GRPO) (Guo et al., 2025) to jointly constrain intermediate reasoning and final clinical decisions. Rather

than rewarding only the correctness of the predicted action, we exploit the multi-granular supervision provided by ORBIT-LD to enforce consistency across perception, temporal reasoning, and treatment planning.

We define a composite reward

$$R_{\text{total}} = w_1 R_{\text{fmt}} + w_2 R_{\text{find}} + w_3 R_{\text{conc}} + w_4 R_{\text{strat}}, \quad (6)$$

where each component addresses a distinct aspect of clinical reliability. $R_{\text{fmt}}$ enforces structural validity of the generated output, supporting stable downstream execution. $R_{\text{find}}$ regularizes biomarker identification using a soft-penalty recall scheme to suppress hallucinations under severe class imbalance. $R_{\text{conc}}$ aligns predicted disease trends with ground-truth longitudinal dynamics, while $R_{\text{strat}}$ applies a hierarchical gated reward to enforce the causal dependency between intervention choice and temporal planning.

### 4.4. Visual Simulator Optimization

We repurpose Qwen-Image-Edit (Wu et al., 2025) to approximate the transition dynamics $\mathcal{T}(s'|s, a)$. To bridge the modality gap, we employ a deterministic projection $\psi(a_t) \rightarrow T_{\text{instr}}$ that translates discrete clinical actions, such as $\langle \text{Inject}, 4 \text{ weeks} \rangle$, into intervention-based instruc-

*Table 1.* **Performance on the Perception Task.** We evaluate multi-label biomarker recognition on the in-domain ORBIT-LD Test split and the out-of-distribution OLIVES dataset. Note that for ORBIT-LD, values denote averages across all 6 biomarkers (DRT/ME, IRF, IR HRF, SHRM, SRF, VMT); for OLIVES, evaluation is restricted to the 3 prevalent biomarkers (DRT/ME, IRF, IR HRF) due to distribution imbalance. **Bold** indicates the best performance.

| Models | ORBIT-LD Test | | | | OLIVES | | | |
|---|---|---|---|---|---|---|---|---|
| | Avg. Acc ↑ | Avg. Rec ↑ | Avg. Jaccard ↑ | Hamm. ↓ | Avg. Acc ↑ | Avg. Rec ↑ | Avg. Jaccard ↑ | Hamm. ↓ |
| *General VLMs* | | | | | | | | |
| Qwen3-VL-8B-Instruct (Team, 2025b) | 64.66 | 37.94 | 36.85 | 35.34 | 49.18 | 26.14 | 25.96 | 50.82 |
| Qwen3-VL-32B-Instruct (Team, 2025b) | 70.59 | 44.92 | 41.63 | 29.41 | 53.55 | 32.54 | 31.78 | 46.56 |
| Qwen2.5VL-72B-Instruct (Bai et al., 2025) | 62.31 | 32.69 | 29.96 | 37.69 | 55.43 | 38.72 | 36.46 | 44.57 |
| InternVL3.5-78B (Wang et al., 2025c) | 68.58 | 41.34 | 40.32 | 31.42 | 61.03 | 43.78 | 42.19 | 38.97 |
| *Medical VLMs* | | | | | | | | |
| Hulu-Med-14B (Jiang et al., 2025) | 69.67 | 44.94 | 43.43 | 31.33 | 62.50 | 41.11 | 40.94 | 37.50 |
| Hulu-Med-32B (Jiang et al., 2025) | 71.70 | 51.77 | 49.16 | 28.30 | 64.99 | 47.53 | 46.67 | 35.01 |
| Lingshu-32B (Xu et al., 2025) | 70.09 | 50.71 | 47.90 | 29.91 | 50.45 | 26.77 | 26.75 | 49.55 |
| QoQ-Med-VL-32B (Dai et al., 2025) | 61.98 | 35.11 | 34.77 | 38.02 | 49.18 | 26.14 | 25.96 | 50.82 |
| **ORBIT (Ours)** | **78.12** | **65.74** | **62.43** | **21.88** | **69.75** | **48.58** | **47.46** | **30.25** |

tions (Wang et al., 2025b; 2024). For instance, the discrete action is mapped to the prompt: "*Simulate the retinal state after anti-VEGF injection with a 4-week interval*".

Crucially, these instructions specify the *intervention* rather than the *outcome*, compelling the model to implicitly learn complex pharmacodynamic responses solely from longitudinal training data.

The training minimizes a hybrid objective:

$$\mathcal{L}_{\text{sim}}(\phi) = \mathbb{E}_{t,\epsilon,I,T}\left[\|\epsilon - \epsilon_\phi(z_t, t, I_t, T_{\text{instr}})\|_2^2\right] \\ + \lambda_{\text{str}}\mathcal{L}_{\text{LPIPS}}(\hat{I}, I_{gt}) \tag{7}$$

The noise prediction term ensures adherence to the treatment condition, while $\mathcal{L}_{\text{LPIPS}}$ (Zhang et al., 2018) enforces biological structure preservation, constraining the model to modify only pathological regions while freezing the rigid retinal topology.

The implementation details and training configurations are provided in Appendix D.

### 4.5. Inference Process

At test time, ORBIT performs decision making via a closed-loop Hypothesize–Simulate–Rectify process that grounds treatment planning in counterfactual anatomical prediction. Given the current observation and history, the policy infers the disease trend and proposes a candidate treatment, which is then evaluated by the learned world model through a forward rollout to predict future anatomy under the proposed intervention.

Instead of committing to this proposal in an open-loop manner, ORBIT reflects on the simulated outcome to assess clinical risk and revises the decision if unresolved pathology or progression is detected. The final output is thus a verified

clinical plan that is consistent with anticipated anatomical evolution rather than a direct policy prediction.

Implementation details and qualitative analyses are provided in Appendix E. In all experiments, the Rectify threshold is fixed to $\tau_{\text{risk}} = 0.7$; Appendix E.3 shows that the closed-loop decision performance remains stable for $\tau_{\text{risk}} \in [0.6, 0.8]$.

## 5. Experiments

### 5.1. Experimental Setup

We evaluate ORBIT within a unified POMDP-grounded framework that encompasses three subtasks: (1) **Perception** (extracting biomarkers), (2) **Evolution** (inferring dynamics), and (3) **Decision** (formulating treatment strategies). ORBIT is trained on the ORBIT-LD training split, and all baselines are evaluated under the same protocol/prompt unless otherwise specified. Additional out-of-distribution assessment is conducted on the public OLIVES (Prabhushankar et al., 2022) dataset for the Perception task. Comprehensive dataset statistics, baseline specifications, and evaluation metrics are detailed in Appendix F.

### 5.2. Perception: Single-Scan Biomarker Recognition

As detailed in Table 1, ORBIT sets a new benchmark in biomarker extraction, achieving **78.12% Accuracy** on the in-domain ORBIT-LD dataset. This performance surpasses not only the leading generalist model but also the domain-specific specialist, *Hulu-Med-32B* (71.70%), by a significant **6.42% margin**. Crucially, ORBIT minimizes the **Hamming Loss to 21.88**, a sharp reduction compared to baselines (e.g., 28.30 for Hulu-Med-32B). This metric indicates that ORBIT achieves precise multi-label calibration directly in the inference pass, effectively avoiding the tendency of other

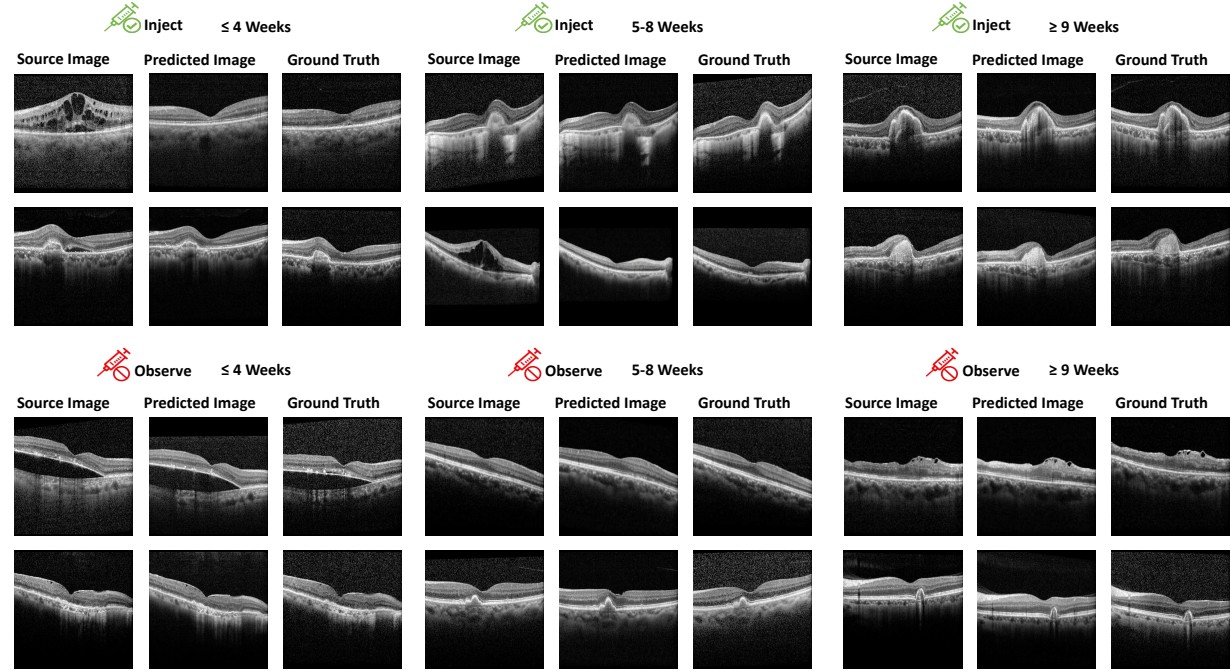

*Figure 4.* **Evaluation of Spatiotemporal Dynamics Fidelity.** Visual comparison highlights the high anatomical consistency between model predictions and Ground Truth. The generator accurately captures complex **pharmacodynamic evolutions**, supporting physical plausibility for downstream planning.

models to hallucinate non-existent comorbidities. Furthermore, in OOD evaluation on OLIVES, ORBIT maintains robust generalization (69.75% Acc), outperforming the 78B-parameter InternVL3.5 by 8.7%, confirming its resilience to scanner heterogeneity. For a fine-grained performance analysis across individual biomarkers and detailed comparisons with other models, please refer to Appendix G.1.

### 5.3. Evolution: Longitudinal Dynamics Modeling

For the Evolution task (Table 2), ORBIT establishes a new state-of-the-art in **Accuracy (80.38%)**. A critical analysis reveals that baseline medical models, such as *Lingshu-32B*, suffer from poor F1 (36.03%), indicating a systemic tendency to misinterpret minor static fluctuations as disease worsening. By explicitly modeling transition dynamics, ORBIT effectively filters these temporal artifacts. This high precision is clinically vital, as it prevents the generation of "false alarms" that would otherwise trigger unnecessary intervention planning for stable patients.

### 5.4. Decision: Treatment Planning

ORBIT achieves the highest **Overall Accuracy (88.52%)** and uniquely overcomes the extreme behavioral biases of baselines (Table 3). Existing models exhibit dangerous polarity: *Qwen3-VL-32B-Instruct* shows an **"interventionist bias"** (99.31% Sens / 11.11% Spec), recommending injection for nearly all patients (over-treatment risk), while

*Table 2.* **Performance on the Evolution Task.** Comparison of disease trend classification (Improved, Stable, Worsened) on longitudinal visit pairs. ORBIT demonstrates superior temporal consistency.

| Models | Acc | Precision | Recall | F1 |
|---|---|---|---|---|
| *General VLMs* | | | | |
| Qwen3-VL-8B-Instruct (Team, 2025b) | 61.20 | 50.87 | 51.70 | 39.14 |
| Qwen3-VL-32B-Instruct (Team, 2025b) | 47.60 | 27.70 | 34.54 | 14.28 |
| Qwen2.5VL-72B-Instruct (Bai et al., 2025) | 58.97 | 46.82 | 51.13 | 36.85 |
| InternVL3.5-78B (Wang et al., 2025c) | 66.56 | 52.72 | **58.49** | 46.81 |
| *Medical VLMs* | | | | |
| Hulu-Med-14B (Jiang et al., 2025) | 55.85 | 43.95 | 46.90 | 32.17 |
| Hulu-Med-32B (Jiang et al., 2025) | 61.20 | 44.79 | 46.97 | 28.63 |
| Lingshu-32B (Xu et al., 2025) | 58.31 | 45.13 | 47.98 | 36.03 |
| QoQ-Med-VL-32B (Dai et al., 2025) | 72.69 | 32.34 | 43.66 | 36.20 |
| **ORBIT (Ours)** | **80.38** | **65.29** | 50.93 | **53.24** |

*QoQ-Med-VL-32B* shows a **"conservative bias"** (26.04% Sens / 88.49% Spec), failing to detect active disease (under-diagnosis risk).

In contrast, ORBIT maintains a balanced decision profile with high **Sensitivity (82.29%)** and **Specificity (95.63%)**. For interval recommendation, ORBIT achieves competitive **Overall Interval Accuracy (54.07%)**, while substantially improving action accuracy and specificity.

### 5.5. Ablation Experiment

We analyze the impact of Visual Foresight and our training strategy on the Decision task (Table 4).

*Table 3.* **Performance on the Decision Task.** Evaluation of clinical management capabilities across First Visit, Follow-up, and Overall settings. Metrics include binary Action Accuracy (Inject vs. Observe), Sensitivity, Specificity, and Interval Accuracy.

| Models | Acc | | | Sens | | | Spec | | | Interval Acc | | |
|---|---|---|---|---|---|---|---|---|---|---|---|---|
| | First | F-up | Overall | First | F-up | Overall | First | F-up | Overall | First | F-up | Overall |
| *General VLMs* | | | | | | | | | | | | |
| Qwen3-VL-8B-Instruct (Team, 2025b) | 79.25 | 69.57 | 73.89 | 90.70 | 77.36 | 83.33 | 66.07 | 60.71 | 63.10 | 51.87 | 43.81 | 47.41 |
| Qwen3-VL-32B-Instruct (Team, 2025b) | 58.09 | 58.19 | 58.15 | **99.22** | **99.37** | **99.31** | 10.71 | 11.43 | 11.11 | 56.02 | **55.52** | **55.74** |
| Qwen2.5VL-72B-Instruct (Bai et al., 2025) | 66.39 | 67.89 | 67.22 | 95.35 | 72.96 | 82.99 | 33.04 | 62.14 | 49.21 | 56.02 | 49.83 | 52.59 |
| InternVL3.5-78B (Wang et al., 2025c) | 75.10 | 73.24 | 74.07 | 95.35 | 70.44 | 81.60 | 51.79 | 76.43 | 65.48 | **56.43** | 45.48 | 50.37 |
| *Medical VLMs* | | | | | | | | | | | | |
| Hulu-Med-14B (Jiang et al., 2025) | 80.91 | 70.90 | 75.37 | 93.02 | 86.16 | 89.24 | 66.96 | 53.57 | 59.52 | **55.19** | 49.50 | 52.04 |
| Hulu-Med-32B (Jiang et al., 2025) | 83.40 | 79.59 | 79.63 | 94.57 | 92.45 | 93.40 | 70.54 | 58.57 | 63.89 | 54.36 | 42.81 | 47.96 |
| Lingshu-32B (Xu et al., 2025) | 73.86 | 73.91 | 73.89 | 96.12 | 95.60 | 95.83 | 48.21 | 49.29 | 48.81 | 54.77 | 29.10 | 40.56 |
| QoQ-Med-VL-32B (Dai et al., 2025) | 54.63 | 54.18 | 55.19 | 23.26 | 28.30 | 26.04 | **94.64** | 83.57 | 88.49 | 20.33 | 31.10 | 26.30 |
| **ORBIT (Ours)** | **87.97** | **88.96** | **88.52** | 82.95 | 81.76 | 82.29 | 93.75 | **97.14** | **95.63** | 54.77 | 53.51 | 54.07 |

*Table 4.* **Ablation study and baseline comparison on the Decision Task.** We evaluate the impact of the Visual Foresight module and training strategies, incorporating AUC (computed on binary predictions) to measure discriminative capability. **C.L.** denotes Closed-Loop architecture, and **Baseline** refers to the generalist Qwen3-VL-8B-Instruct model.

| METHOD | C.L. | ACC | AUC | SENS | SPEC |
|---|---|---|---|---|---|
| BASELINE | × | 73.89 | 72.32 | 83.33 | 63.10 |
| SFT ONLY | × | 86.30 | 86.41 | 84.72 | 88.10 |
| RL ONLY | × | 87.96 | 87.85 | 89.58 | 86.11 |
| SFT + RL | × | 86.85 | 86.73 | 88.54 | 84.92 |
| **ORBIT** | ✓ | **88.52** | **88.96** | 82.29 | **95.63** |

**Impact of Visual Foresight.** First, the generalist baseline (Qwen3-VL) struggles with this specialized task (72.32% AUC), underscoring the necessity of domain-specific optimization. Furthermore, the closed-loop architecture significantly outperforms open-loop counterparts. Incorporating visual foresight boosts the **AUC to 88.96%** (vs. 86.73% for the Open-Loop SFT+RL) while achieving the highest accuracy. This confirms that counterfactual simulation provides superior anatomical grounding, enabling robust decisions compared to pure text reasoning.

**Impact of Training Strategy.** The results highlight the efficacy of the combined strategy. While RL alone improves sensitivity, the holistic **SFT+RL** approach in ORBIT achieves the best comprehensive performance. It attains the highest **Accuracy (88.52%)** and **Specificity (95.63%)**, effectively balancing the precision-risk trade-off by encouraging necessary intervention while minimizing the socioeconomic burden of over-treatment.

### 5.6. Visual Foresight Analysis

**Quantitative Analysis.** We first evaluate the Visual Simulator on real longitudinal OCT pairs. Compared with the orig-

inal Qwen-Image-Edit backbone, the fine-tuned simulator substantially improves perceptual and distributional fidelity, reducing LPIPS from 0.8400 to 0.4018 and DINOv3-FID from 37.62 to 2.761. The simulated futures also reach a biomarker agreement of 0.8121 with real follow-up images, supporting their use as clinically meaningful verification signals. Full metrics are provided in Appendix D.5.

**Qualitative Analysis.** To validate the interpretability and physical grounding of ORBIT, we visualize the internal predictive dynamics of the World Model. Our analysis focuses on two critical dimensions: the fidelity of disease simulation and the validity of counterfactual reasoning.

We first evaluate the spatiotemporal dynamics fidelity as presented in Fig. 4. Leveraging the World Model as a patient-specific "Digital Twin," we condition predictions on source images and ground-truth clinical instructions. The qualitative results demonstrate high anatomical consistency between model predictions and real follow-up scans. Crucially, the model accurately captures complex pharmacodynamic evolutions—synthesizing fluid absorption and PED flattening in injection cases while preserving lesion persistence in observation cases. This confirms that the model minimizes hallucinations and ensures the physical plausibility required for downstream planning.

Subsequently, we verify the model's capacity for counterfactual prognosis across the full joint action space (Fig. 5). The results demonstrate that the model produces distinct anatomical outcomes for the six treatment variants, indicating strong sensitivity to intervention prompts and effectively avoiding mode collapse. Moreover, the ground-truth clinical plans (highlighted by the red boxes) consistently correspond to the most favorable visual prognoses among the generated counterfactuals. In high-risk cases (e.g., Case C), the expert-selected intervention aligns with the largest degree of fluid resolution, whereas in stable cases (e.g., Case A), the expert

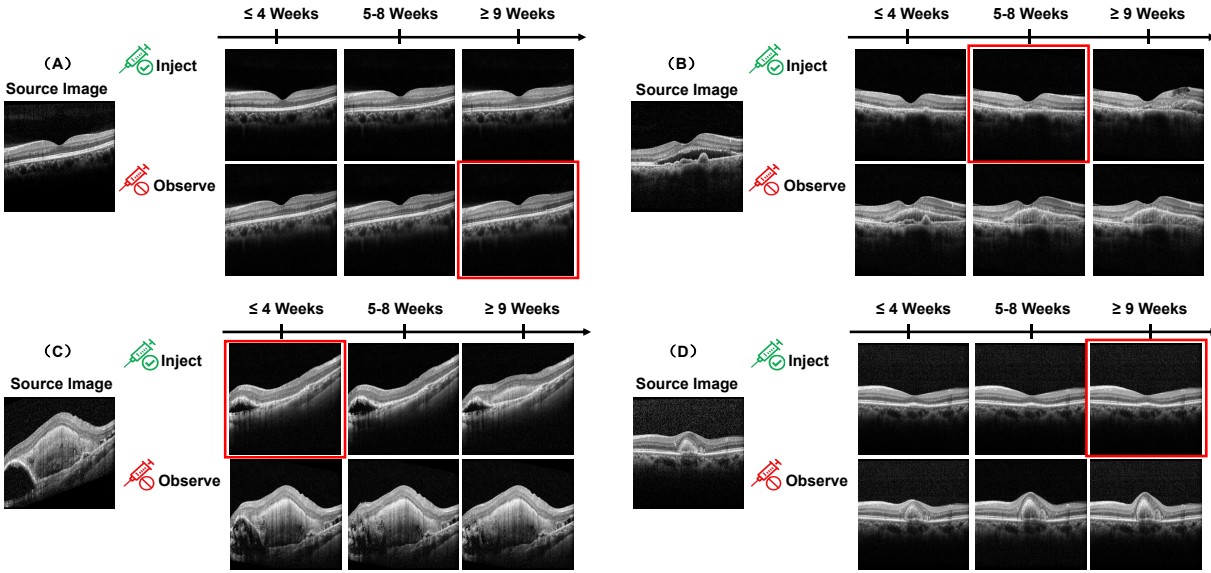

*Figure 5.* **Counterfactual Prognosis in Joint Action Space.** We demonstrate model **sensitivity** to intervention prompts, resulting in distinct anatomical outcomes across the six treatment variations. Crucially, **Red Boxes** (expert decisions) consistently align with the **optimal visual prognoses**, validating the correct causal mapping from clinical policy to physical outcome.

choice achieves comparable anatomical stability to more aggressive interventions while minimizing treatment burden. These observations confirm that the model intrinsically captures the causal relationship between clinical policies and their resulting physical outcomes.

## 6. Conclusion and Limitations

**Conclusion.** In this work, we present **ORBIT**, a prognostic world model for longitudinal ophthalmic disease management. To mitigate the supervision gap in clinical archives, we introduce a Logic-Constrained Abductive Data Engine that derives dense pathological trajectories from sparse clinical records. By incorporating counterfactual visual foresight into the decision loop, ORBIT enables candidate actions to be evaluated through simulated anatomical evolution and verification, providing a structured alternative to purely open-loop text-based reasoning. Our results suggest that this formulation can improve decision-support performance while offering a more interpretable framework for human-in-the-loop ophthalmic management.

**Limitations.** ORBIT is intended as a step toward human-in-the-loop longitudinal decision support rather than an autonomous clinical decision maker. Its closed-loop inference introduces additional test-time computation, reflecting a trade-off between efficiency and counterfactual verification. Although the verification gates help reduce explicit decision–biomarker inconsistencies and implausible temporal transitions, residual biases from the data engine, base MLLM, or training distribution may remain. In addition, the current

simulator produces a single representative rollout without calibrated patient-specific uncertainty estimates, and the use of central B-scans may miss rare off-foveal abnormalities. Broader validation across scanners, institutions, patient populations, and treatment protocols is therefore needed before considering real-world clinical deployment.

## Acknowledgements

This work was conducted during Jiangtao Yan's internship at the Shanghai Artificial Intelligence Laboratory. This work was supported by the Shanghai Artificial Intelligence Laboratory and the National Natural Science Foundation of China under Grant No. 62571379 and 62376200.

## Impact Statement

This work explores prognostic world modeling for longitudinal ophthalmic decision support. By incorporating counterfactual visual simulation into the decision loop, ORBIT may help clinicians compare potential anatomical outcomes under different treatment strategies and support more transparent human-in-the-loop reasoning. This could help reduce unnecessary interventions and patient burden if validated in prospective clinical settings. However, ORBIT is not intended to replace clinical judgment or serve as an autonomous decision-making system, and its predictions may reflect biases in training data, imaging protocols, or model-generated supervision. Careful validation across institutions, scanners, patient populations, and treatment practices is therefore required before real-world deployment.

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

# A. Logic-Constrained Abductive Data Engine

This section provides the implementation specifications and design rationale for the **Logic-Constrained Abductive Data Engine** introduced in Sec. 3. The primary objective of this engine is to resolve the *supervision gap* inherent in longitudinal clinical archives: while therapeutic actions ($y_t$) are explicitly recorded, the granular pathological states ($s_t$) motivating these decisions are typically unannotated. To address this, our approach adopts a "Propose-and-Verify" paradigm rather than direct state reconstruction. Specifically, we leverage a Multimodal LLM to first propose candidate pathological states based on visual evidence, and subsequently employ clinical outcomes and temporal continuity as logical premises to rigorously verify the plausibility of these candidates. This process ensures that the finalized disease trajectories are biologically plausible and temporally coherent, effectively recovering dense pathological information from sparse supervision signals.

## A.1. Data Ethics Statement

This study utilizes the ORBIT-LD dataset, derived from retrospective ophthalmic archives at our institution. All data collection, processing, and usage were approved by the Institutional Review Board (IRB), with a waiver of informed consent granted due to the retrospective and non-interventional nature of the research. We strictly adhered to relevant privacy protection regulations throughout the data processing pipeline, ensuring the removal of all Protected Health Information (PHI) and the implementation of irreversible de-identification procedures.

## A.2. Algorithmic Framework

Algorithm 1 summarizes the full data construction pipeline. The engine operates in three stages that directly correspond to the modules described in Sec. 3: **Hierarchical Semantic Synthesis (Propose)**, **Dual-Consistency Verification (Verify)**, and **Trajectory Modeling (Label)**. This distinct separation ensures that potential hallucinations produced by the generative VLM are strictly filtered by deterministic clinical logic, thereby improving data quality.

### A.3. Module 1: Hierarchical Semantic Synthesis (Propose)

Given the characteristics of OCT imaging, we process the data as a sequence of multiple 2D slices rather than a single volumetric 3D block. Directly feeding all slices into a VLM would incur prohibitive computational costs and introduce redundant information. Therefore, this module adopts a hierarchical processing strategy. Following the convention of OCT clinical diagnosis, we select five slices at the foveal center to represent the overall anatomical situation as local visual evidence. Subsequently, an **Aggregation Module** summarizes and reconciles these slice-level descriptions, eliminating redundancy and resolving semantic conflicts. This results in the synthesis of a unified candidate biomarker set $\mathcal{S}_t$, covering six key pathological indicators (e.g., IRF, SRF, DRT/ME, SHRM, IR HRF, VMT), alongside a preliminary structured report. This step constitutes the "Propose" phase of the data engine.

### A.4. Module 2: Dual-Consistency Verification (Verify)

To ensure the biological fidelity of the candidate reports, the generated biomarker set $\mathcal{S}_t$ undergoes rigorous logical gating. This process directly utilizes clinical decisions as validation conditions to assess the rationality of the LLM's generation.

The **Decision-Biomarker Consistency Gate** enforces causal alignment between pathology and treatment. Specifically, an injection decision ($y_t = $ Inject) clinically implies the presence of active disease. Consequently, if a report generated for an injection sample lacks any biomarkers (i.e., $\mathcal{S}_t = \emptyset$), this directly indicates that the generative model has hallucinated a False Negative. Such samples are marked as invalid and trigger regeneration. Conversely, for observation samples ($y_t = $ Observe) where biomarkers are present, we route them to expert review to distinguish stable residual pathology from errors, rather than simply rejecting them.

Subsequently, the **Temporal Plausibility Gate** further regularizes the trajectory based on the gradual nature of retinal disease evolution. Rather than treating all temporal changes as invalid, the gate identifies high-risk biomarker substitution patterns, where at least one biomarker present at visit $t - 1$ disappears and at least one different biomarker newly emerges at visit $t$:

$$(S_{t-1} \setminus S_t \neq \emptyset) \wedge (S_t \setminus S_{t-1} \neq \emptyset). \tag{8}$$

These cases are routed for human-in-the-loop review to distinguish genuine rapid progression or treatment response from MLLM generation errors. In addition, the gate flags large changes in the total biomarker burden,

$$\left| |S_t| - |S_{t-1}| \right| \geq 3, \tag{9}$$

for further review. Through this verification mechanism grounded in logical premises, we improve the temporal coherence and clinical plausibility of the finalized dataset.

## A.5. Module 3: Trajectory Modeling

Following the verification steps, the sequence enters the final trajectory modeling phase. Here, a **Pairwise Comparator** computes the qualitative trend of disease evolution (Improved, Stable, or Worsened) by contrasting the verified report at the current time $t$ with that of the previous time $t-1$. This generates the dense evolution labels $l_t$ required for training the world model's transition dynamics, completing the transformation from sparse clinical records to dense pathological trajectories.

## A.6. Quantitative Analysis of Verification Gates

To assess the reliability of the Logic-Constrained Abductive Data Engine, we conduct a manual audit on 540 generated samples. The audit focuses on whether the verification gates intercept high-risk errors introduced during the MLLM-based proposal stage. Table 5 summarizes the results.

The audit shows that the dual verification gates reduce two major categories of high-risk errors: decision–biomarker inconsistency and implausible temporal evolution. Since deterministic constraints cannot eliminate subtle or systematic biases that do not explicitly violate predefined logic, expert review is retained as an additional safeguard.

---

**Algorithm 1** Logic-Constrained Abductive Data Engine

---

**Require:** Longitudinal patient sequence $\mathcal{D} = \{(I_t, y_t)\}_{t=1}^{T}$, where $I_t$ is a 3D OCT volume and $y_t \in \{\text{Inject}, \text{Observe}\}$
**Ensure:** Verified reports $\mathcal{R} = \{R_1, \ldots, R_T\}$ and evolution labels $\mathcal{L} = \{l_2, \ldots, l_T\}$
 1: // Phase 1: Hierarchical Semantic Synthesis
 2: **for** $t = 1$ **to** $T$ **do**
 3:      Generate slice-level visual descriptions from $I_t$ using a multimodal LLM
 4:      Aggregate and reconcile descriptions to obtain a structured report $\hat{R}_t$ and biomarker set $\mathcal{S}_t$
 5:      // Phase 2: Dual-Consistency Verification
 6:      **if** $y_t = \text{Inject}$ **and** $\mathcal{S}_t = \emptyset$ **then**
 7:          Mark $\hat{R}_t$ as Invalid and trigger regeneration
 8:      **else if** $y_t = \text{Observe}$ **and** $\mathcal{S}_t \neq \emptyset$ **then**
 9:          Route $\hat{R}_t$ to expert review
10:          **if** generation or annotation error is confirmed **then**
11:              Trigger regeneration
12:          **end if**
13:      **end if**
14:      **if** $t > 1$ **then**
15:          **if** $(\mathcal{S}_{t-1} \setminus \mathcal{S}_t \neq \emptyset) \wedge (\mathcal{S}_t \setminus \mathcal{S}_{t-1} \neq \emptyset)$ **or** $\big| |\mathcal{S}_t| - |\mathcal{S}_{t-1}| \big| \geq 3$ **then**
16:              Route $\hat{R}_t$ to expert review
17:              **if** generation or annotation error is confirmed **then**
18:                  Trigger regeneration
19:              **end if**
20:          **end if**
21:      **end if**
22:      $R_t \leftarrow \hat{R}_t$ {Accept verified report}
23: **end for**
24: // Phase 3: Trajectory Modeling
25: **for** $t = 2$ **to** $T$ **do**
26:      Compare $R_{t-1}$ and $R_t$ using a Pairwise Comparator
27:      Assign evolution label $l_t \in \{\text{Improved}, \text{Stable}, \text{Worsened}\}$
28: **end for**
29: **return** $\mathcal{R}, \mathcal{L}$

---

*Table 5.* Quantitative audit of the verification gates on 540 samples. The Decision–Biomarker Consistency Gate captures inconsistencies between clinical actions and proposed biomarkers, while the Temporal Plausibility Gate captures biologically implausible longitudinal transitions.

| Stage | Trigger condition | Action | Count |
|---|---|---|---|
| Propose | Candidate report and biomarker set | Proceed to validation | 540 |
| Decision–Biomarker Consistency Gate | Inject but no biomarker is proposed | Reject and regenerate | 29 |
| Decision–Biomarker Consistency Gate | Observe but biomarker is present | Route to expert review | 123 |
| Expert Review | High-risk observation samples | Confirm hallucination or error | 56 |
| Temporal Plausibility Gate | Implausible simultaneous change | Review or regenerate | 35 pairs |
| Temporal Plausibility Gate | Abrupt biomarker-burden shift | Review or regenerate | 13 pairs |

*Table 6.* Comparison of ORBIT-LD with representative public OCT datasets. ORBIT-LD uniquely combines longitudinal OCT acquisitions, dense biomarker annotations, and explicit clinical action sequences, supporting decision-making and world-modeling tasks under a POMDP formulation. Scale denotes dataset size as reported in the original publications.

| Dataset | Modality | Scale | Longitudinal | Clinical Action | Biomarker | Primary Task |
|---|---|---|---|---|---|---|
| OCT2017 | 2D OCT | ~84k | ✗ | ✗ | Category | Classification |
| OCT-C8 | 2D OCT | ~24k | ✗ | ✗ | Category | Multi-class Cls. |
| OCTDL (Kulyabin et al., 2024) | 2D OCT | ~1.6k | ✗ | ✗ | Category | Multi-class Cls. |
| OCTA-500 (Li et al., 2024) | OCT + OCTA | 500 | ✗ | ✗ | ✓ | Segmentation |
| OLIVES (Prabhushankar et al., 2022) | OCT + IR | 96 eyes | ✓ | Partial | ✓ | Biomarker Detection |
| **ORBIT-LD (Ours)** | **Multi-slice OCT** | **2,969 seqs / 5,435 visits** | ✓ | ✓ | ✓ | **Decision / World Model** |

## B. Dataset Statistics and Clinical Validity Analysis

This appendix provides a detailed analysis of the ORBIT-LD dataset. We first compare it with existing public ophthalmic datasets to elucidate its uniqueness in supporting longitudinal decision-making tasks (Sec. B.1). Subsequently, we demonstrate the clinical realism and annotation quality through statistical distribution analysis and causal logic verification (Sec. B.2), followed by qualitative case studies (Sec. B.3).

### B.1. Comparison with Existing Public Datasets

Existing public ophthalmic datasets have significantly advanced the field of static automated diagnosis. However, they remain fundamentally constrained by their cross-sectional nature, rendering them insufficient for training agentic models capable of longitudinal reasoning. As detailed in Table 6, we compare ORBIT-LD with representative datasets. Mainstream datasets such as **OCT2017** and **Retinal OCT-C8** primarily focus on static classification tasks; while providing large-scale 2D imagery, their annotations are limited to discrete disease categories (e.g., CNV vs. Normal), lacking the temporal dimension required to model disease progression over time. Even datasets offering fine-grained pathological segmentation or multi-modal imaging, such as **OCTDL** (Kulyabin et al., 2024) and **OCTA-500** (Li et al., 2024), do not encode clinical interventions (e.g., Anti-VEGF injection records) and thus cannot support the learning of counterfactual treatment policies. Although the **OLIVES** (Prabhushankar et al., 2022) dataset introduces longitudinal scans, it is limited in scale and does not explicitly formulate the *Action-State-Reward* tuples necessary for POMDP modeling. In contrast, **ORBIT-LD** is uniquely constructed to bridge this gap. By coupling dense biomarker annotations with longitudinal clinical actions ($a_t$) and evolution labels ($y_{evol}$), it serves as the first dedicated dataset for Prognostic World Models and Offline Reinforcement Learning in ophthalmology.

### B.2. Statistical Properties and Clinical Validity

To validate ORBIT-LD as a robust dataset for world models, we conduct a comprehensive analysis covering distributional characteristics, anatomical consistency of abductive labeling, and latent causal dynamics.

**Longitudinal Heterogeneity and Distribution.** ORBIT-LD faithfully reflects the complexity of real-world clinical environments. The ORBIT-LD training split exhibits a pronounced long-tailed distribution in sequence lengths (**Fig. 6A**, $N = 2728$), encompassing extended management trajectories suitable for learning long-term temporal dependencies. The swimmer plot (**Fig. 6B**) further visualizes the temporal irregularity of patient follow-ups and the interleaving of "Inject/Observe" actions. Regarding distribution balance, the dataset maintains a relatively balanced action space (Inject 42.7% vs. Observe 57.3%, **Fig. 6C1**) and diverse recommended follow-up intervals (**Fig. 6C2**, with < 4 weeks and > 8 weeks being comparable). Additionally, the trend distribution (**Fig. 6C3**) indicates that the majority of follow-ups are stable

(55.7%), consistent with chronic disease management.

**Validation of Abductive Logic.** Since our pathological labels are generated via abductive reasoning, validating their biological plausibility is rigorous. Clinical Logic Verification (**Fig. 6F**) demonstrates that our engine successfully captures the causal premises of therapeutic decisions: the probability of injection rises significantly ($> 0.7$) when key biomarkers such as Intraretinal Fluid (IRF) or Subretinal Fluid (SRF) are present, whereas it remains low ($\approx 0.2$) in their absence. Furthermore, the Anatomical Correlations heatmap (**Fig. 6E**) reveals rational co-occurrence relationships. For instance, Diffuse Retinal Thickening (DRT/ME) exhibits a strong positive correlation with IRF ($\rho = 0.81$), aligning perfectly with the pathophysiology of retinal edema.

**Causal Dynamics and Policy Response.** The dataset encodes clear "Intervention-Outcome" dynamics. State Transition Matrices indicate that distinct policies lead to divergent prognoses: the "Inject" action significantly promotes recovery ($P(\text{Improved}|\text{Worsened}) = 0.71$, **Fig. 6D2**), while the "Observe" policy is primarily associated with state persistence (**Fig. 6D1**). Concurrently, the expert policy demonstrates dynamic adaptability (**Fig. 6G**), with injection rates peaking at 68% during the "Worsened" phase to control pathology. This efficacy is validated by the Longitudinal Evolution (**Fig. 6H**), where the proportion of "Stable" patients expands progressively (from 33% at Visit 1 to 70% at Visit 5) over the treatment course.

### B.3. Qualitative Longitudinal Case Studies

To visually substantiate the dataset's microscopic structure, Fig. 7 and Fig. 8 present representative longitudinal OCT case studies. These examples explicitly illustrate the temporal coherence between pathology changes (e.g., fluid resolution or recurrence) and clinical actions (Inject or Observe). By demonstrating anatomically plausible evolutions across multiple visits, these visual evidences further validate the temporal consistency and clinical rationality of ORBIT-LD at the individual granularity.

## C. Policy Learning Implementation Details

This appendix details the hyperparameter configurations and data synthesis protocols for the optimization curriculum outlined in Section 4.3.

### C.1. Structured SFT: From Discrete Labels to CoT Serialization

While the ORBIT-LD dataset provides rigorously validated pathological annotations, these labels exist in a **discrete and fragmented** state, which do not inherently constitute a reasoning chain. To bridge the gap between these scattered evidence points and coherent clinical logic, we implement a **Structured Serialization Protocol**.

This mechanism functions as a logic compiler. We employ Gemini 2.5-Pro (Comanici et al., 2025) to act as a "Senior Retinal Specialist," transforming the ground-truth tuple $(O_t, h_{t-1}, Y_{bio}, Y_{evol}, Y_{action})$ into a fluent, causal narrative sequence. Unlike generic templates, our prompt engine enforces a strict **three-step reasoning paradigm** that dynamically adapts to the patient's visit type. Finally, we concatenate the generated treatment rationale with the discrete annotations to form a complete Chain-of-Thought (CoT) sequence.

The specific prompt template is illustrated below:

---

**Prompt Template: Expert Treatment Rationale Generation**

# Role
You are a senior retinal ophthalmology specialist. Your task is to write a logically rigorous and professional **"### 3. Treatment Rationale"** based on the patient's **[Basic Information]**, **[Medical History]**, **[Current Imaging Features]**, **[Disease Progression Trend]**, and **[Final Treatment Decision]**.

# Input Data
Please read the following standardized fields:

1. **Patient Overview**: Basic information: {basic_info}; Clinical diagnosis: {clinical_text}; Visit type: {visit_type} (values: "Initial Visit" or "Follow-up Visit").

2. **Medical History (only for follow-up)**: Previous treatment: {history_treatment} (focus on: interval since last injection); Historical comparison: {trend_status}; Rationale: {trend_reasoning} (changes from $S_0$ vs $S_1$).

3. **Current Imaging Features**: Findings: {findings}; Impression: {impression}.

4. **Final Decision**: Operation: {operation} (Inject/Observe); Follow-up: {follow_up_weeks}.

# Reasoning Logic (Step-by-Step)
Please write strictly following the three-step logic below. Note that you must choose different reasoning pathways depending on {{visit_type}}.

### 1. Correlation Analysis

- **Task**: Link the "etiology" with the "morphology."

- **General logic**: Combine the primary disease in {{clinical_text}} (e.g., DR, RVO) with morphological features in {{findings}} (e.g., IRF, SRF). Explain that anatomical changes are caused by **breakdown of the blood–retinal barrier**, **ischemia/hypoxia**, or **inflammatory leakage**.

### 2. Priority & Activity

- **Task**: Determine disease activity and urgency. Select the pathway based on {{visit_type}}:

- **Path A: [Initial Visit (Baseline)]**

  - Ignore historical information. Directly assess severity. If terms like "diffuse" or "massive" appear, emphasize **"the disease is in the initial acute phase, requiring urgent intervention."**

- **Path B: [Follow-up Visit]**

  - Incorporate {{history_treatment}} (interval) and {{trend_status}}.
  - **If Worsened**: Analyze interval. If long ($> 8$ weeks), state **"recurrence due to drug washout."** If short, state **"high disease activity refractory to standard therapy."**
  - **If Stable/Improved**: State **"regimen is effective, lesions in absorption phase."**

### 3. Clear Conclusion

- **Task**: Provide final justification for {{operation}} and {{follow_up_weeks}}.

- **Operation**: *Injection* (suppress VEGF based on acute/worsening status) vs. *Observation* (stable).

- **Follow-up**: *Shortened* ($\leq 4$ *weeks*) (close monitoring for worsening/recurrence) vs. *Routine* (stable).

# Output Format
Only output the section **"### 3. Treatment Rationale"** in Markdown format.

---

## C.2. Holistic Process-Outcome RL (Hyperparameter Configuration)

In the GRPO phase, the composite reward function $R_{\text{total}}$ employs a uniform weighting scheme ($w_1 = w_2 = w_3 = w_4 = 1$) to maintain equilibrium across syntactic, semantic, and strategic objectives.

**1) Format Compliance ($R_{\mathbf{fmt}}$).** To guarantee parsing validity, structural constraints are enforced with weights $\gamma = [0.2, 0.2, 0.6]$, prioritizing JSON integrity as a critical engineering constraint:

$$R_{\text{fmt}} = 0.2 \cdot \mathbb{I}(\text{Tag}_{\text{perc}}) + 0.2 \cdot \mathbb{I}(\text{Tag}_{\text{conc}}) + 0.6 \cdot \mathbb{I}(\text{Valid}_{\text{json}}) \tag{10}$$

**2) Biomarker Identification ($R_{\text{find}}$).** A Soft-Penalty Recall mechanism is applied with penalty coefficient $\lambda = 0.2$. This threshold was empirically determined to balance sensitivity against hallucination suppression, particularly for rare pathological features:

$$R_{\text{find}} = \frac{|\hat{Y}_{\text{bio}} \cap Y_{\text{bio}}| + \epsilon}{|Y_{\text{bio}}| + \epsilon} - 0.2 \cdot |\hat{Y}_{\text{bio}} \setminus Y_{\text{bio}}| \tag{11}$$

**3) Conclusion Consistency ($R_{\text{conc}}$).** Logical consistency is enforced via binary alignment with the longitudinal ground truth ($y_{\text{evol}}^{\text{gt}}$), masked by history availability ($M_{\text{hist}}$) to prevent noise during initial visits:

$$R_{\text{conc}} = M_{\text{hist}} \cdot \mathbb{I}(\hat{y}_{\text{evol}} = y_{\text{evol}}^{\text{gt}}) \tag{12}$$

**4) Therapeutic Strategy ($R_{\text{strat}}$).** The Hierarchical Gated Reward utilizes a base coefficient $\alpha = 0.6$, allocating 60% of the reward mass to the correct intervention type (Inject/Observe) and 40% to temporal precision:

$$R_{\text{strat}} = \mathbb{I}(\hat{a}_{\text{op}} = y_{\text{op}}) \cdot [0.6 + 0.4 \cdot \mathbb{I}(\hat{a}_{\text{time}} = y_{\text{time}})] \tag{13}$$

## D. Visual Simulator Optimization Details

We detail the optimization of the Visual Simulator ($f_\phi$), instantiated via **Qwen-Image-Edit**.

### D.1. Weakly-Supervised Implicit Learning Paradigm

A distinguishing feature of our framework is its adoption of a **weakly-supervised implicit learning paradigm**. Crucially, the training protocol obviates the need for explicit supervision of anatomical changes (e.g., difference maps or deformation fields) and labor-intensive pixel-level segmentation masks.

Instead, the model is presented solely with pre- and post-intervention image pairs ($I_t, I_{t+1}$) conditioned on the treatment action $a_t$ (injected via the instruction $T_{instr}$). By minimizing the reconstruction objective, the model is compelled to autonomously induce the causal pharmacodynamic effects triggered by specific interventions (e.g., learning that anti-VEGF injections correlate with fluid resorption). This design significantly reduces the annotation burden for large-scale longitudinal datasets.

### D.2. Instruction Template Construction

To condition the diffusion process, discrete clinical actions are projected into natural language prompts. The action space $\mathcal{A}$ comprises two dimensions: **Intervention** ($\{\text{Anti-VEGF Injection}, \text{Observation}\}$) and **Interval** ($\{\leq 4 \text{ weeks}, 5\text{-}8 \text{ weeks}, > 8 \text{ weeks}\}$). These components are composed into structured prompts $T_{instr}$ (e.g., *"Simulate the retinal state after anti-VEGF injection with a 4-week interval"*), effectively bridging the modality gap between clinical decisions and visual synthesis.

### D.3. Dataset Statistics

The Visual Simulator is trained on a separate transition corpus for image-level prognostic modeling. This corpus contains 11,982 visit-to-visit OCT transitions ($I_t, I_{t+1}$). For each transition, we extract five central B-scans centered at the fovea, yielding 59,910 image-level transition pairs. This corpus is used only for optimizing the Visual Simulator and is distinct from the ORBIT-LD policy-learning split described in Appendix F.1.

### D.4. Training Configuration

Training leverages the **DiffSynth-Studio** framework. The model is initialized with Qwen-Image-Edit weights, with fine-tuning applied to the denoising backbone and attention layers to adapt the pretrained generative prior to ophthalmic feature spaces. The objective function minimizes the reconstruction error of $I_{t+1}$ conditioned on $I_t$ and $T_{\text{instr}}$, allowing the model to internalize causal intervention effects from longitudinal image pairs without explicit anatomical change supervision.

### D.5. Quantitative Evaluation of the Visual Simulator

In addition to the qualitative results, we quantitatively evaluate the Visual Simulator on a held-out set of real longitudinal OCT image pairs. We compare the fine-tuned simulator with the original Qwen-Image-Edit backbone using pixel-level

*Table 7.* Full quantitative evaluation of the Visual Simulator on real longitudinal OCT pairs. SSIM and PSNR measure low-level fidelity, LPIPS measures perceptual similarity, DINOv3-FID and DINOv3-KID measure feature-distribution consistency, and biomarker agreement measures clinical semantic consistency.

| Metric | Qwen-Image-Edit | Visual Simulator |
|---|---|---|
| SSIM $\uparrow$ | 0.1445 | 0.3556 |
| PSNR $\uparrow$ | 9.82 | 24.32 |
| LPIPS $\downarrow$ | 0.8400 | 0.4018 |
| DINOv3-FID $\downarrow$ | 37.62 | 2.761 |
| DINOv3-KID ($\times 1000$) $\downarrow$ | 72.50 | 3.27 |
| Biomarker agreement $\uparrow$ | – | 0.8121 |

---

**Algorithm 2** Hypothesize–Simulate–Rectify Inference Pipeline (Test-Time)

---

1: **Input:** Observation $O_t$, History $h_{t-1}$, Policy $\pi_\theta$, World Model $f_\phi$, Threshold $\tau_{risk}$
2: **Output:** Verified Clinical Plan $a_{final}$
3: **Hypothesize (Perception & Evolution)**
4: $\quad s_{perc} \leftarrow \pi_\theta(\text{Perception} \mid O_t)$
5: $\quad s_{evol} \leftarrow \pi_\theta(\text{Evolution} \mid s_{perc}, h_{t-1})$
6: $\quad a_{hyp} \sim \pi_\theta(\text{Foresight} \mid s_{evol})$
7: **Simulate (Visual Rollout)**
8: $\quad \hat{I}_{next} \leftarrow f_\phi(O_t, a_{hyp})$
9: **Rectify (Safety Reflection)**
10: $\quad s_{refl} \leftarrow \pi_\theta(\text{Reflection} \mid a_{hyp}, \hat{I}_{next})$
11: $\quad r_{safety} \leftarrow \text{EvaluateRisk}(s_{refl})$
12: **if** $r_{safety} > \tau_{risk}$ **then**
13: $\quad\quad a_{final} \leftarrow \text{HeuristicAdjust}(a_{hyp}, \text{Conservative})$
14: **else**
15: $\quad\quad a_{final} \leftarrow a_{hyp}$
16: **end if**
17: **return** $a_{final}$

---

fidelity, perceptual similarity, and feature-distribution metrics. We also report biomarker agreement, computed by comparing biomarkers detected from simulated future images with those from corresponding real follow-up images. Results are shown in Table 7.

The fine-tuned Visual Simulator consistently improves fidelity and distributional metrics over the original image-editing backbone. The biomarker agreement score further suggests that the generated futures preserve clinically meaningful semantic changes. Since the unadapted backbone does not reliably satisfy OCT anatomical constraints, biomarker agreement is reported only for the fine-tuned simulator.

## E. Closed-Loop Inference and Safety Verification

This appendix details the closed-loop inference mechanism employed by ORBIT at test time, including the formal definition of the inference algorithm, the safety reflection and risk evaluation strategy, and representative qualitative inference case studies. This section complements the high-level description in the main text and provides full algorithmic clarity and interpretability.

### E.1. Inference Algorithm: Hypothesize–Simulate–Rectify

Algorithm 2 summarizes the Hypothesize–Simulate–Rectify inference pipeline used by ORBIT during test time. The algorithm tightly couples the policy network with the learned world model, validating candidate treatment plans through counterfactual anatomical prediction rather than relying on a single open-loop forward pass.

By explicitly introducing counterfactual simulation and safety reflection stages, this inference procedure ensures that final

*Table 8.* Sensitivity analysis of the Rectify threshold $\tau_{\text{risk}}$. ORBIT remains stable within 0.6–0.8, while more extreme thresholds show the expected sensitivity-specificity trade-off.

| $\tau_{\text{risk}}$ | Acc | Sens | Spec |
|---|---|---|---|
| 0.4 | 83.70 | 86.46 | 80.56 |
| 0.6 | 88.33 | 82.99 | 94.44 |
| 0.7 | 88.52 | 82.29 | 95.63 |
| 0.8 | 88.15 | 80.90 | 96.43 |
| 0.9 | 78.15 | 59.72 | 99.21 |

decisions are grounded in verifiable future anatomical trajectories rather than a single forward prediction of the policy network.

### E.2. Safety Reflection and Risk Evaluation

During the Rectify stage, the policy network assumes the role of a safety critic, evaluating the clinical risk associated with the counterfactual future state generated by the world model. Risk assessment focuses on whether pathological activity is adequately mitigated under the proposed treatment or whether signs of persistence or progression remain evident. If the estimated risk exceeds a predefined threshold $\tau_{risk}$, the system triggers a model-based conservative adjustment to revise the candidate plan and prevent potentially unsafe clinical decisions.

This safety reflection mechanism enables ORBIT to automatically detect and intervene on high-risk decisions without relying on manually specified rules, improving robustness and reliability in real-world clinical settings. In all main experiments, we set $\tau_{\text{risk}} = 0.7$ and analyze its sensitivity in Appendix E.3.

### E.3. Threshold Sensitivity Analysis

The Rectify stage triggers decision revision when the estimated risk score exceeds the threshold $\tau_{\text{risk}}$. To examine whether ORBIT is sensitive to this hyperparameter, we evaluate decision performance under different threshold values. Results are shown in Table 8.

ORBIT remains stable for $\tau_{\text{risk}} \in [0.6, 0.8]$, suggesting that the Rectify mechanism does not rely on fine-grained threshold tuning. Lower thresholds improve sensitivity at the cost of specificity, whereas overly high thresholds substantially reduce sensitivity. We therefore use $\tau_{\text{risk}} = 0.7$ as a balanced default.

### E.4. Qualitative Inference Case Studies

To validate the robustness of the "Hypothesize-Simulate-Rectify" inference pipeline, we present two representative real-world cases where the closed-loop architecture successfully averted potential under-diagnosis errors (i.e., "False Negatives" in intervention). In both scenarios, due to the ambiguity of static biomarkers, the initial open-loop policy ($\pi_\theta$) favored a conservative "Observation" strategy. However, the counterfactual foresight of the World Model revealed the risk of disease progression, thereby triggering our model to rectify the final decision.

As shown in Figure 10, the patient presented with parafoveal focal Subretinal Hyperreflective Material (SHRM) and localized Subretinal Fluid (SRF). The Policy Agent initially interpreted the limited fluid volume and the absence of significant Intraretinal Fluid (IRF) as signs of low disease activity, thus proposing a conservative strategy of "Observation". However, when simulating the consequences of this "Observation" action over a future 5–8 week horizon, the Visual Simulator ($\hat{I}_{next}$) predicted a marked exacerbation of pathological features, specifically manifesting as a significant increase in macular SRF forming a dome-shaped elevation. Upon analyzing this generated prognostic image, the Reflection module detected that the simulated outcome contradicted the goal of disease stability. Consequently, it overruled the initial hypothesis, reasoning that "Increased SRF indicates high activity; observation is insufficient," and our model subsequently rectified the final decision to "Intravitreal Injection".

Figure 11 illustrates a more challenging case involving a high myopic patient presenting with an irregular mass-like lesion. The Agent initially misclassified the lesion as a "chronic, stable scar" due to its dense SHRM appearance, deeming the mild SRF as not indicative of active leakage, and thus recommended "Observation". The World Model subsequently executed

a rollout conditioned on "Observation", predicting a dramatic expansion of the SRF extent and an increase in SHRM volume after 8 weeks. This visual evidence effectively served as a "stress test" for the initial diagnosis. Confronted with the visualized worsening trend, the system recognized the active neovascular nature of the lesion rather than stability. Our model immediately rectified the decision to "Intravitreal Injection" and shortened the follow-up interval to within 4 weeks to control the rapid progression.

These cases highlight the unique value of ORBIT compared to standard medical VLMs: while a standard VLM might halt at an initial plausible-sounding but clinically dangerous text output, ORBIT leverages the World Model as a causal verifier. By grounding decisions in future anatomical consequences rather than merely current semantic probabilities, it provides a critical safety layer for human-in-the-loop decision support.

## F. Experimental Setup Details

### F.1. Dataset Protocols

**ORBIT-LD Statistics.** ORBIT-LD is partitioned at the longitudinal-sequence level into training and held-out test splits, ensuring that all visits from the same sequence are assigned to a single split. The training split contains 4,895 policy-learning samples, including 2,728 first-visit samples and 2,167 follow-up samples, while the held-out ORBIT-LD Test split contains 540 samples, including 241 first-visit samples and 299 follow-up samples. In total, ORBIT-LD contains 5,435 policy-learning samples from 2,969 longitudinal sequences and 2,466 follow-up visits. Since each sequence starts with one first visit, the number of first-visit samples equals the number of longitudinal sequences. The held-out test split is used exclusively for final evaluation.

**OLIVES (OOD).** For Out-of-Distribution evaluation, we utilize the public OLIVES dataset. Incorporating the clinical prior of foveal dominance, we sample the three central B-scans from each OCT volume, resulting in a test set of **1,071 samples**. *Note on Biomarker Selection:* Due to the highly imbalanced distribution of biomarkers in the OLIVES dataset, we restricted our quantitative evaluation exclusively to the three prevalent biomarkers: **DRT/ME, IRF, and IR HRF**.

### F.2. Definitions of Evaluation Metrics

We evaluate model performance from three complementary aspects: biomarker perception, disease evolution modeling, and treatment decision planning.

For biomarker perception, we evaluate multi-label recognition at the patient-visit level. For each visit $i$, the model predicts a biomarker set $\hat{\mathcal{Y}}_i$, which is directly compared with the ground-truth set $\mathcal{Y}_i$. This patient-visit-level formulation evaluates the complete biomarker pattern within each sample, jointly accounting for correct detections, false positives, and missed biomarkers. We report Accuracy, Recall, average Jaccard index, and Hamming Loss. The sample-level Jaccard index measures the overlap between the predicted and ground-truth biomarker sets:

$$\text{Jaccard}_i = \frac{|\hat{\mathcal{Y}}_i \cap \mathcal{Y}_i|}{|\hat{\mathcal{Y}}_i \cup \mathcal{Y}_i|}. \tag{14}$$

The sample-level Recall measures the proportion of ground-truth biomarkers that are correctly detected:

$$\text{Recall}_i = \frac{|\hat{\mathcal{Y}}_i \cap \mathcal{Y}_i|}{|\mathcal{Y}_i|}. \tag{15}$$

Hamming Loss is computed over all sample–label pairs:

$$\text{HL} = \frac{1}{NL} \sum_{i=1}^{N} \sum_{l=1}^{L} \mathbb{I}\left[\hat{y}_{i,l} \neq y_{i,l}\right], \tag{16}$$

where $N$ is the number of visits, $L$ is the number of evaluated biomarkers, and $\hat{y}_{i,l}, y_{i,l} \in \{0, 1\}$ denote the predicted and ground-truth labels for biomarker $l$ at visit $i$, respectively. Evaluation is conducted over six biomarkers on ORBIT-LD, including IRF, SRF, DRT/ME, SHRM, IR HRF, and VMT, and over three selected biomarkers on OLIVES, including DRT/ME, IRF, and IR HRF. Lower Hamming Loss indicates fewer label-wise prediction errors.

For disease evolution modeling, we formulate visit-to-visit disease dynamics as a three-class classification task, where each transition is categorized as *Improved*, *Stable*, or *Worsened*. We report Accuracy, Precision, Recall, and F1 Score. Accuracy

reflects the overall correctness of trend prediction, Precision measures the reliability of predicted trend categories, Recall measures the ability to identify actual trend instances, and F1 Score summarizes the trade-off between Precision and Recall.

For treatment decision planning, we evaluate two clinically relevant sub-tasks: binary treatment action prediction, i.e., *Inject* versus *Observe*, and follow-up interval recommendation. Results are reported under **First Visit**, **Follow-up Visit**, and **Overall** settings to distinguish initial decision-making from longitudinal management. For action prediction, we report Accuracy, Sensitivity, and Specificity. Sensitivity corresponds to the recall of the *Inject* class and reflects the ability to identify patients requiring anti-VEGF treatment, whereas Specificity corresponds to the recall of the *Observe* class and reflects the ability to avoid unnecessary treatment. For interval recommendation, we report Interval Accuracy, which measures whether the predicted follow-up interval matches the doctor-recommended interval.

### F.3. Ablation on the Number of Central B-scans

We further examine the effect of the number of central B-scans used as visual input. Although full OCT volumes provide richer anatomical coverage, macular lesions are typically concentrated near the foveal center, and using excessive slices may introduce redundant computation under the current model capacity. Table 9 reports a preliminary ablation over different numbers of central slices.

The results suggest that too few slices may miss local anatomical variation, while too many slices can introduce redundant information. This ablation does not imply that full-volume OCT modeling is unnecessary; rather, it supports the use of five central B-scans as a practical and efficient setting for the present implementation.

*Table 9.* Ablation study on the number of central B-scans used as visual input. Using five central B-scans achieves the best overall performance, suggesting a favorable balance between complementary information and redundant slices.

| # Central B-scans | Acc. | F1 |
|:---:|:---:|:---:|
| 1 | 0.741 | 0.772 |
| 2 | 0.750 | 0.775 |
| 3 | 0.748 | 0.776 |
| 4 | 0.772 | 0.796 |
| **5** | **0.778** | **0.801** |
| 6 | 0.756 | 0.783 |
| 7 | 0.746 | 0.774 |
| 8 | 0.719 | 0.752 |

## G. Additional Experimental Results

### G.1. Fine-grained Biomarker Analysis

To rigorously evaluate model performance across distinct pathological features, we present fine-grained performance radar charts comparing ORBIT with baseline models on both the in-domain ORBIT-LD and out-of-distribution OLIVES datasets (Fig. 9).

As illustrated, ORBIT (red star) demonstrates comprehensive superiority, encompassing the largest closed area across all biomarker axes in both datasets. Notably, even for challenging and rare lesions such as **Vitreomacular Traction (VMT)**, ORBIT maintains high sensitivity and specificity. In contrast, other generalist VLMs and medical-specific models (e.g., Hulu-Med-32B) exhibit significant performance imbalance: while they achieve competitive results on common biomarkers like DRT/ME or IRF, they suffer from severe performance collapse on complex lesions. This consistent equilibrium attests to ORBIT's capability to provide holistic and reliable diagnostic support, demonstrating robustness even in complex cases with multiple comorbidities.

---

**Test Prompt: Ophthalmic Expert AI**

**1. Role & Core Task**

You are a world-class Ophthalmic Expert specializing in retinal diseases and OCT analysis. Your task is to generate a diagnostic assessment and treatment plan based on clinical text (history, complaints) and OCT images (focusing on the 5 "Current Status" slices and comparing with "History" if available).

---

**2. Analysis Framework**
- **Biomarker Detection:** Carefully scan for and analyze the following. *If found, detail morphology/location in "impression" and list names in "findings".*
  - **Targets: VMT** (Traction), **IRF** (Intraretinal Fluid - note shape/layer), **DRT/ME** (Edema), **IR HRF**, **SRF** (Subretinal Fluid - note shape/loc), **SHRM**.
- **Priority Logic: Current OCT activity determines treatment.** If active exudation (IRF, SRF) is present, prioritize aggressive treatment (Injection) over observation, regardless of historical trends.

---

**3. Output Template (Strictly follow this structure)**

**Assessment & Recommendations**

**1. Current Status Assessment**
 **impression:** {Comprehensive impression. Describe OCT morphology/lesions. Detail abnormal biomarkers here.}
 **findings:** {Python list of abbreviations found, e.g., `["IRF", "SRF"]` or `[]` if none.}

**2. Evaluation of Disease Progression**
 **Evaluation Process:** {Compare fluid/edema with history. If first visit, state "First visit, no data for comparison".}
 **conclusion:** {Select one: **Improved / Stable / Worsened / N/A**}

**3. Treatment Rationale** 1. **Correlation:** {Link history (Diabetes/RVO) with current fluid/edema.}
 2. **Priority:** {How current OCT activity guided the decision.}
 3. **Conclusion:** {Explicit reason for Injection vs. Observation.}

**4. Treatment Strategy Recommendation**

Output a standard JSON block. **No markdown text inside JSON.**
- `follow_up_weeks`: Choose **"<= 4 weeks"**, **"5-8 weeks"**, or **">= 9 weeks"**.

```json
{
    "operation":  "Intravitreal Injection",
    "follow_up_weeks":  "<= 4 weeks"
}
```

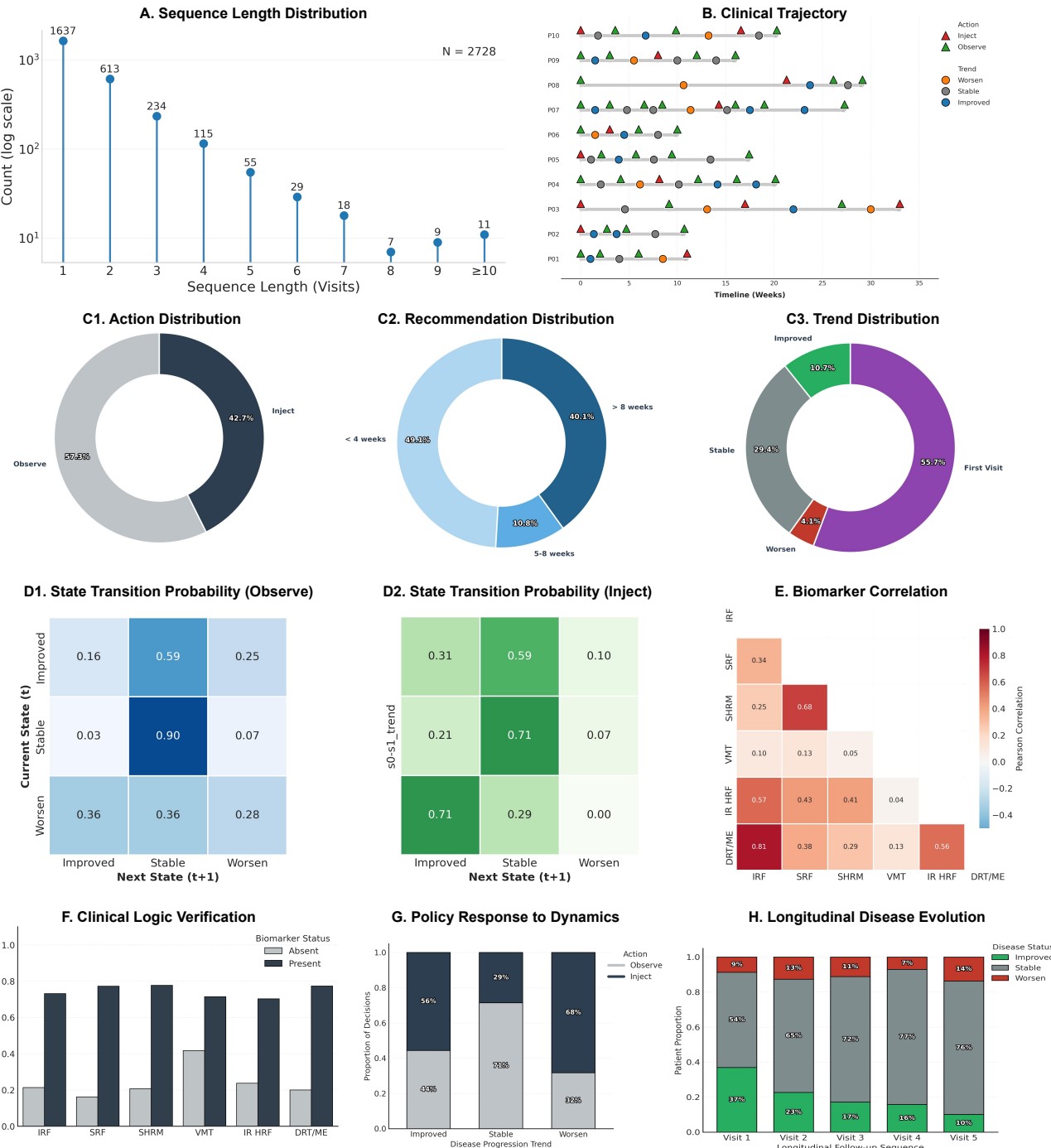

*Figure 6.* **Statistical Analysis and Clinical Verification of the Longitudinal Dataset. (A)** The sequence length distribution of the ORBIT-LD training split exhibits a long-tailed pattern ($N = 2728$), reflecting the variance in patient follow-up durations. **(B)** Representative swimmer plots visualize the temporal heterogeneity of clinical trajectories, interleaving actions (Inject/Observe) with evolving disease states. **(C1–C3)** Global statistics reveal a balanced action space (Inject: $42.7\%$) and realistic pathological diversity across visits. **(D1–D2)** State transition matrices quantify the intervention dynamics within the POMDP: while the "Observe" policy favors state persistence ($P(\text{Stable}|\text{Stable}) = 0.90$), the "Inject" policy significantly drives recovery from worsening states ($P(\text{Improved}|\text{Worsened}) = 0.71$). **(E)** Biomarker correlation heatmap confirms anatomical consistency, highlighted by the strong correlation between fluid presence (IRF) and retinal thickening (DRT/ME) ($\rho = 0.81$). **(F)** Clinical logic verification validates the **Abductive Data Engine**, demonstrating that injection probability is explicitly conditioned on actionable biomarkers (e.g., IRF, SRF). **(G)** Policy response analysis reveals expert adaptability: injection rates rise to $68\%$ during disease worsening phases while maintaining a maintenance dose ("Treat-and-Extend") during improvement. **(H)** Longitudinal cohort evolution across five visits demonstrates the effectiveness of the encoded strategies in stabilizing disease progression over time.

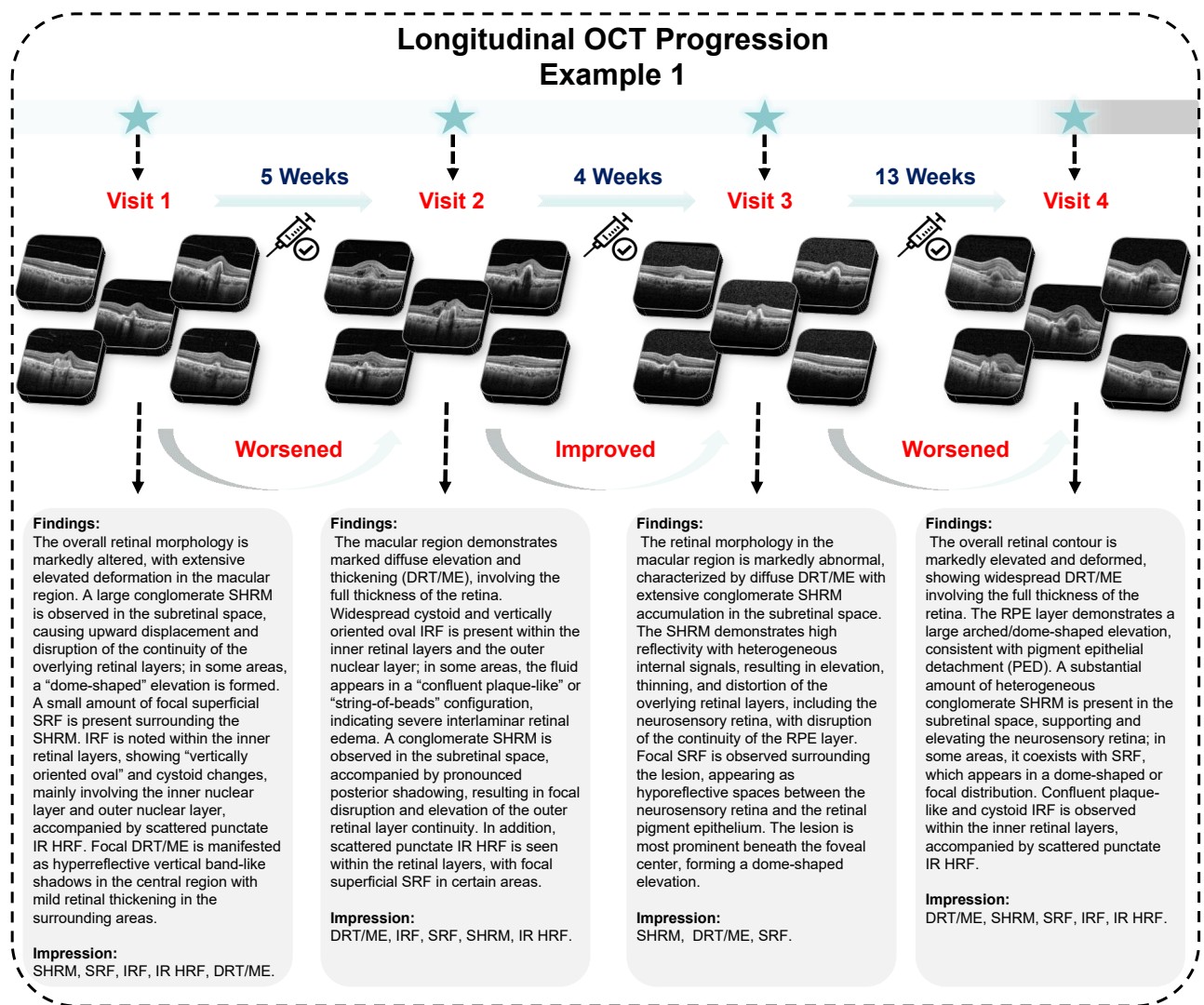

*Figure 7.* **Longitudinal OCT progression example 1**. A longitudinal case illustrating a multi-visit OCT sequence with variable inter-visit intervals and recorded clinical interventions. The example contains individual visit-level OCT scans, their temporal ordering, and explicit state transitions (worsened and improved) across visits.

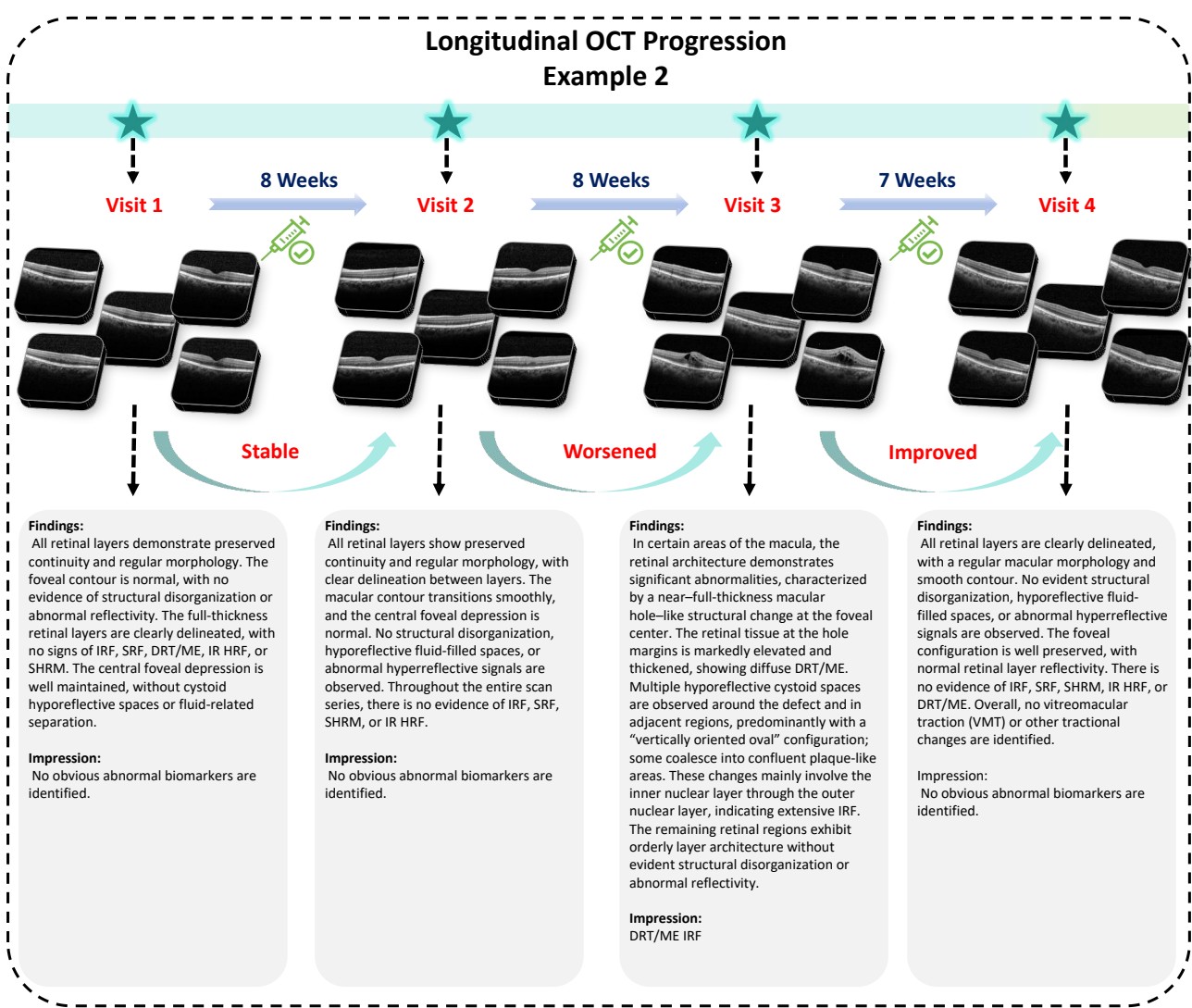

*Figure 8.* **Longitudinal OCT progression example 2**. Another longitudinal case demonstrating the dataset structure with single-visit OCT records, sequential visit information, and annotated inter-visit state changes (stable, worsened, and improved) over time.

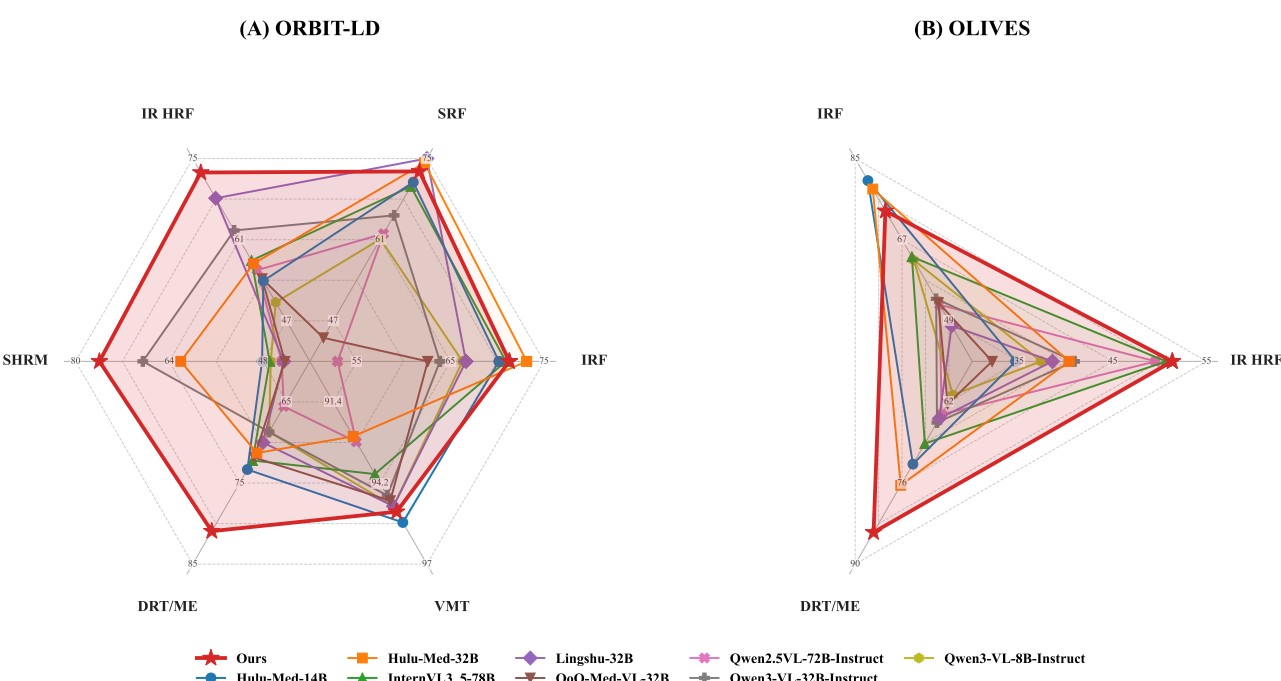

*Figure 9.* Fine-grained biomarker performance comparison. We visualize the performance radar charts on (A) ORBIT-LD (In-Domain) and (B) OLIVES (Out-of-Distribution) datasets. ORBIT (Red Star) demonstrates superior and balanced performance across all pathological categories.

---

**Case Study: Interactive Diagnosis & Treatment Loop**

**1. Clinical Input (Visit: 2024-03-13, First Visit, Right Eye)**
- **History:** Chief complaint of blurred vision (> 2 months). Dx: Senile cataract.
- **Goal:** Provide treatment recommendations based on OCT & History.

**Current OCT Images (Baseline):**

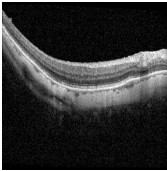 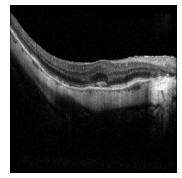 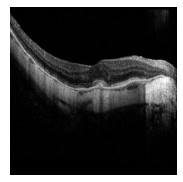 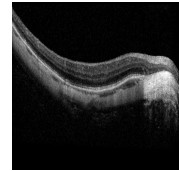 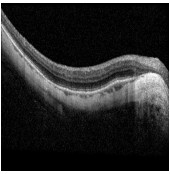

---

**2. Assessment & Recommendation**

**1. Current OCT Assessment:**
**Impression:** Retinal structure is largely preserved. Focal **SHRM** is noted parafoveally (slightly right of the fovea), appearing as dense hyperreflective material above the RPE/subretinal space with mild outer retinal elevation/disorganization. Localized **SRF** is present in the macular/parafoveal area. No significant **IRF** or **DRT/ME** is observed.
**Findings:** SRF, SHRM

**2. Disease Change:** First visit; no prior data for comparison.

**3. Treatment Suggestion:** Given limited SRF, dense SHRM (possibly chronic/organized), and absence of IRF/DRT suggesting low activity, **Observation** is recommended. Repeat OCT within **5-8 weeks** to monitor SRF amount and SHRM changes.

```
<CALL_SIMULATOR> Action:  Observation, Follow-up:  5-8 weeks
```

---

**3. Decision Verification & Final Recommendation**

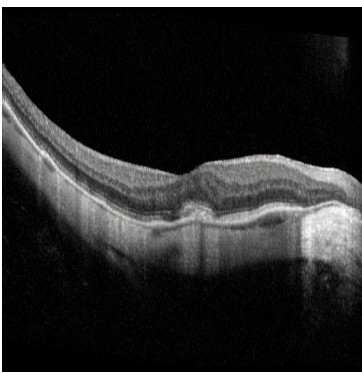

*Simulated Prognosis (5–8 weeks)*

**1. Reflection on Simulation Results:**
- **Visual Feedback:** Macular **SRF** significantly increased (dome-shaped elevation). **SHRM** unchanged but elevated.
- **Conclusion:** Increased SRF indicates high activity; observation is insufficient. Initiate **anti-VEGF injection**.

**2. Final Treatment Strategy:**

```
{
    "operation":  "Intravitreal Injection",
    "follow_up_weeks":  "5-8 weeks"
}
```

*Figure 10.* Interactive Case Study. The model initially suggests observation (Section 2). However, after the simulator predicts significant fluid worsening (Section 3, left image), the model reflects on this visual feedback and corrects the strategy to Intravitreal Injection (Section 3, right text).

## Case Study: Interactive Diagnosis & Treatment Loop

**1. Clinical Input (Visit: 2021-01-12, First Visit, Right Eye)**
- **Patient Info:** Female, Approx. 57 years old.
- **History:** High myopia, corneal epithelial injury.
- **Goal:** Provide treatment recommendations based on OCT & History.

**Current OCT Images (Baseline):**

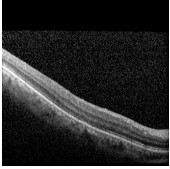 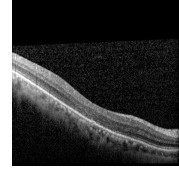 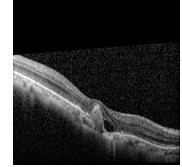 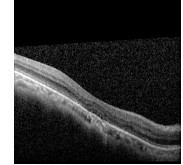 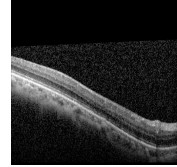

### 2. Assessment & Recommendation

**1. Current OCT Assessment:**
**Impression:** Localized lesion in macular area with marked retinal elevation. Irregular mass-like **SHRM** in subretinal space accompanied by obvious **SRF**, showing a dome-shaped appearance.
**Findings:** SRF, SHRM
**2. Disease Change:** First visit; no prior data for comparison.
**3. Treatment Suggestion:** Macular elevation with dense SHRM suggests a chronic, stable scar; mild SRF may not indicate active leakage. **Observation** is recommended. Repeat OCT in **5-8 weeks**.

```
<CALL_SIMULATOR> Action:  Observation, Follow-up:  5-8 weeks
```

### 3. Decision Verification & Final Recommendation

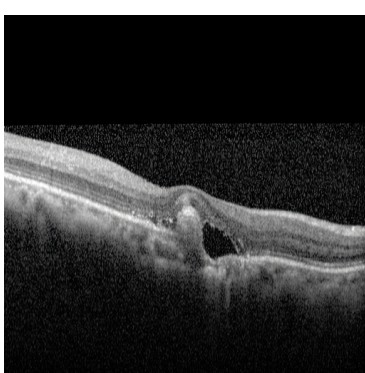

*Simulated Prognosis (5-8 weeks)*

**1. Reflection on Simulation Results:**
- **Visual Feedback: SRF** extent expanded significantly; **SHRM** volume increased.
- **Conclusion:** Marked increase in SRF indicates worsening activity. Observation failed. Initiate **anti-VEGF injection** immediately.

**2. Final Treatment Strategy:**

```
{
    "operation":  "Intravitreal Injection",
    "follow_up_weeks":  "<= 4 weeks"
}
```

*Figure 11.* Interactive Case Study (Patient: Female, 57y). The model initially classifies the lesion as a stable scar (Section 2) and suggests observation. However, the simulator predicts significant fluid expansion (Section 3, left image), prompting an immediate correction to Intravitreal Injection.

