# OpenReview forum: "ORBIT: A Prognostic World Model for Ocular Reasoning Based on Imagined Trajectories"
_ICML.cc/2026/Conference — ICML 2026 regular_

### Official Review · Reviewer_JWE7 · 2026-02-25

**Soundness:** 2
**Presentation:** 3
**Significance:** 2
**Originality:** 2
**Overall Recommendation:** 3
**Confidence:** 3

**Summary:**

This paper introduces ORBIT, a Prognostic World Model designed for longitudinal management of ophthalmic diseases via counterfactual simulation and closed-loop clinical reasoning. Building on a newly constructed ORBIT-LD dataset, the approach leverages a Logic-Constrained Abductive Data Engine to infer dense biomarker trajectories from sparse clinical archives by fusing LLM-based proposal generation and logical consistency filters.

**Compliance With Llm Reviewing Policy:**

Affirmed.

**Final Justification:**

Please see my Rebuttal Acknowledgement

**Key Questions For Authors:**

(1) The dataset is not accompanied by any commitment to future public release, so I do not consider this dataset to be a meaningful contribution of the paper.

(2) From a technical standpoint, the paper’s contributions are fairly limited, mainly combining existing models in a straightforward way. For example, the “Counterfactual Visual Foresight” and COT ideas it highlights are already quite common in the EHR area.

(3) The paper emphasizes clinical significance quite strongly, but properly assessing that (especially the dataset construction and assessment) would require ophthalmology expertise. Given the incremental technical advance and heavy clinical focus, it might be a better fit for venues like MICCAI or Nature Biomedical Engineering than ICML. I’m not entirely sure it aligns well with ICML’s scope.

(4) In real clinical practice, doctors do indeed focus their attention heavily on the macular fovea. However, most medical personnel (including myself) typically scroll through the entire 3D OCT volume, and in some cases also check the sagittal and coronal views. Therefore, I still have concerns about selecting only 5 key frames, especially for certain corner cases where full 3D data has a clear advantage over sparse 2D slices.

(5) Metrics should also be calculated for the visual simulator to demonstrate its accuracy, rather than relying solely on visualization comparisons.

**Limitations:**

No, the authors only emphasize the positive impacts. As a medical study, obvious risks clearly exist.

*Authors should be rewarded rather than punished for being upfront about the limitations of their work and any potential negative societal impact.*

**Strengths And Weaknesses:**

This paper investigates a highly specialized problem in ophthalmology and constructs a dedicated dataset for it. The research is thorough, with substantial data and effort invested. While the problem is important within its niche field, its impact is narrow and limited to this specific organ, offering no particularly novel theoretical contributions. Its main theoretical value lies in formalizing the problem and developing an architecture tailored to this clinical context. See the Key Questions section for details.

---

> ### Author Rebuttal · Authors · 2026-03-31
>
> Q1:
>
> We acknowledge concerns about data availability and reproducibility. Due to privacy and institutional constraints, we cannot commit to a full public release of the raw clinical data. However, we will release as much as permitted, including the logic rules, pseudo-label pipeline, and benchmark protocol, to improve reproducibility and future comparison.
>
> More importantly, the main contribution is a data construction and task formulation paradigm for sparse-supervision longitudinal decision problems, rather than a standalone organ-specific dataset.
>
> ---
>
> Q2：
>
> First, this work is not a simple combination of existing models. Rather than proposing a new backbone, we introduce ORBIT, a future-grounded closed-loop decision framework that integrates partial observability, trajectory reconstruction, intervention-conditioned visual transition, and decision verification. Its key novelty lies in the hypothesize-simulate-rectify loop, where predicted future feedback is used to verify and refine current decisions.
>
> Second, our CoT and counterfactual visual foresight differ fundamentally from prior EHR work. Typical EHR methods operate on tabular or textual states and rely on outcome prediction or textual consistency. In contrast, we model intervention-conditioned future anatomical chang and use predicted future observations / future anatomy to verify current actions. Therefore, our method is not open-loop reasoning, but future-grounded closed-loop decision verification.
>
> | Dimension | EHR Counterfactual / CoT | ORBIT |
> |---|---|---|
> | Counterfactual target | Risk / event / outcome prediction | Future anatomical change under intervention |
> | Verification signal | Outcome prediction / textual consistency | Predicted future observations |
> | Decision style | Open-loop reasoning | Closed-loop hypothesize-simulate-rectify |
>
> ---
>
> Q3：
>
> We understand the reviewer's concern regarding fit with ICML. We clarify that ICML does not exclude medical research, and several recent accepted papers have been medically oriented[1,2]. In addition, this submission was explicitly filed under **Applications→Health/Medicine**, which is a designated area within ICML’s scope.
>
> More importantly, this work is not only about ophthalmology, but about a broader ML problem: grounded decision-making under partial observability with sparse process supervision. We propose a future-grounded closed-loop framework that combines state reconstruction, action-conditioned visual transition modeling, and decision verification for long-horizon decision problems where observations are incomplete, supervision is sparse, and actions affect future states.
>
> In this sense, ophthalmology serves as a structured testbed, rather than the boundary of the contribution. We therefore believe the core contribution is nota domain-specific application, but is a framework for verifiable reasoning about current decisions.
>
> [1] MedRAX: Medical Reasoning Agent for Chest X-ray
>
> [2] Visual and Domain Knowledge for Professional-level Graph-of-Thought Medical Reasoning
>
> ---
>
> Q4:
>
> Our slice-based design does not imply that 3D information is unimportant. Under the current setup, raw 3D OCT volumes are also typically processed as multiple 2D slices, while macular lesions usually occupy only a small portion of the full volume. Using many more slices therefore substantially increases computation while introducing redundancy.
>
> To justify the slice-number choice, we conducted a preliminary ablation around the macular center using a fine-tuned DINOv3-based model. The results suggest that one slice is insufficient, while too many slices introduce redundancy; under the current setting, 5 clinically focused slices provide the best trade-off.
>
> | Nums | Acc | F1 |
> |---|---|---|
> | 1 | 0.741 | 0.772 |
> | 2 | 0.750 | 0.775 |
> | 3 | 0.748 | 0.776 |
> | 4 | 0.772 | 0.796 |
> | 5 | **0.778** | **0.801** |
> | 6 | 0.756 | 0.783 |
> | 7 | 0.746 | 0.774 |
> | 8 | 0.719 | 0.752 |
>
> ---
>
> Q5:
>
> We agree that quantitative evaluation is necessary and have added experiments on a test set of real longitudinal image pairs.
>
> The results show that, compared with the original Qwen-Image-Edit, the fine-tuned Visual Simulator outperforms in both plausibility and quantitative metrics (fidelity and distributional consistency). We also introduced a biomarker-based clinical semantic evaluation, but it was not conducted for Qwen-Image-Edit because its outputs don't meet OCT anatomical requirements.
>
> | Metric | Qwen-Image-Edit | Visual Simulator |
> |---|---|---|
> | SSIM ↑ | 0.1445 | 0.3556 |
> | PSNR ↑ | 9.82 | 24.32 |
> | LPIPS ↓ | 0.8400 | 0.4018 |
> | DINOv3-FID ↓ | 37.62 | 2.761 |
> | DINOv3-KID (×1000) ↓ | 72.50 | 3.27 |
> | Biomarker agreement ↑ | - | 0.8121 |
>
> ---
>
> limitation:
>
> We will add a discussion of limitations and risks, including incomplete raw data release, constraints of sparse-slice inputs versus full-volume OCT, and the system’s role strictly as a human-in-the-loop decision-support tool rather than for autonomous clinical use.

---

> > ### Author Rebuttal · Reviewer_JWE7 · 2026-04-02
> >
> > Thanks to the authors for the clarifications, which successfully addressed a portion of my questions. I will update my score from 2 to 3 accordingly. That being said, the technical novelty of the framework alone does not fully justify a 4 in my assessment. If the authors commit to releasing the dataset and ensuring it is of sufficiently high quality, I would consider increasing my score to a 4.

---

> > > ### Author Response · Authors · 2026-04-03
> > >
> > > Thank you for the careful reading of our rebuttal, and the score update.
> > >
> > >
> > > Regarding data availability, due to clinical privacy and institutional regulations, releasing the raw data requires a formal approval process that can be lengthy and is not fully under our control, so we cannot commit to immediate release. However, once approval is granted, we will release the full dataset without delay. At the same time, we will release all reproducible components upon paper publication, including the logic rules, data construction pipeline, and evaluation protocols, to support future research and fair comparison.
> > >
> > >
> > > In terms of technical contribution, we would like to further clarify that ORBIT is not primarily aimed at introducing a new standalone module, but rather explores a world model-based closed-loop decision formulation (hypothesize–simulate–rectify) for longitudinal clinical reasoning. Compared to typical open-loop approaches, the framework uses simulated intervention-conditioned future states to help assess current decisions, forming a “propose–verify–refine” process. In addition, ORBIT considers intervention-conditioned anatomical trajectory modeling and integrates perception, evolution, and decision within a POMDP framework.

---

### Official Review · Reviewer_NMYT · 2026-03-09

**Soundness:** 3
**Presentation:** 4
**Significance:** 3
**Originality:** 3
**Overall Recommendation:** 4
**Confidence:** 2

**Summary:**

The paper proposes ORBIT, a novel Prognostic World Model to address the critical challenge of balancing timely intervention (e.g., anti-VEGF injections) with the risks of over-treatment. ORBIT simulates future anatomical states (visual foresight) under different counterfactual treatment plans, allowing the model to ground its clinical decisions in simulated biological outcomes rather than relying on the unsafe, open-loop text reasoning typical of standard VLMs.

**Compliance With Llm Reviewing Policy:**

Affirmed.

**Key Questions For Authors:**

1. Biological responses to treatments like anti-VEGF injections exhibit inter-patient variability. How does the visual foresight module in ORBIT account for the stochastic nature of disease progression? Does it generate a single deterministic future state, or can it model a distribution of possible outcomes?
2. In the "Hypothesize-Simulate-Rectify" pipeline, what happens if the initial perception step misses a crucial, subtle biomarker? Can the visual simulator spontaneously generate the correct evolution of a feature that was not explicitly identified in the text hypothesis, or is it strictly constrained by the initial textual perception?
3. Can you discuss the interpretability of the model? Since this is important in real-world applications for AI in health.

**Limitations:**

yes

**Strengths And Weaknesses:**

**Strengths**
1. The paper is well-structured, clearly contrasting traditional vision models, standard VLMs, and the proposed ORBIT framework.
2. Employing visual foresight to validate textual hypotheses provides a grounding mechanism that mitigates VLM hallucinations.

**Weaknesses**
1. The paper states it uses a world model to simulate future states, but it is not entirely clear how the model handles the inherent stochasticity of patient responses to treatment. Biological systems are not perfectly deterministic, and simulating a single "imagined trajectory" might result in false confidence if the variance of the generated outcome is not quantified.
2. The reliance on an LLM to "propose" initial biomarkers and treatments implies that if the initial VLM proposal is completely out of distribution, the visual verification step might struggle to correct it entirely.

---

> ### Author Rebuttal · Authors · 2026-03-31
>
> Q1:
>
> We appreciate the reviewer for highlighting the inherent stochasticity in patient responses to treatments. As the reviewer correctly points out, patient responses to treatments inherently exhibit stochastic variability, and medical prognosis is not fully deterministic. The design goal of ORBIT is to enable **risk comparison across candidate interventions**, rather than to precisely predict a single future state.
>
> In the current implementation, ORBIT does not perform uncertainty-aware distribution modeling. Instead, it generates a single representative rollout for each candidate intervention, which is used for comparative verification across actions. Figure 5 in our paper illustrates how this mechanism produces corresponding future anatomical states under different treatment strategies.
>
> Importantly, this design does not imply that we assume disease progression is deterministic. What's more, explicit modeling of uncertainty under the same intervention can be naturally extended via multi-sample rollouts, which is an important direction for future work.
>
> In the revision, we will clarify that the current simulator serves as a single-sample counterfactual verifier, rather than a fully uncertainty-aware generative forecaster.
>
>
> ---
>
> Q2:
>
> We thank the reviewer for this important question.  We first clarify that the visual simulator is **not conditioned on textual descriptions of lesion features**. Instead, it simulates disease progression based on the **current image state and intervention instruction**.
>
> If the initial perception step misses a subtle but important biomarker, the policy model may incorrectly assess the current condition as stable and recommend a low-risk strategy such as *observation*. In such a case, the visual simulator, starting from the **current state combined with the suboptimal intervention**, may generate a **future state with pathological deterioration**.
>
> Therefore, the simulator can reveal **inconsistencies between the planned intervention and the predicted anatomical evolution**, even when the initial perception is imperfect.
>
> ---
>
>
> Q3:
>
>
> The interpretability of the proposed system operates at three levels:
>
> 1. **Structured perception / biomarker report**
> 2. **Explicit action hypothesis**
> 3. **Simulated anatomical consequences used for verification**
>
> Additionally, during RL training, rewards are assigned not only to final outcomes but also to intermediate reasoning processes such as biomarker identification.
>
> We emphasize that the simulated futures generated by the visual simulator should be viewed as **decision-support signals**. While interpretability does not guarantee absolute correctness, compared to “black-box” or “open-loop” reasoning, our framework provides a **more inspectable and transparent reasoning process for clinicians**.

---

> > ### Author Rebuttal · Reviewer_NMYT · 2026-04-03
> >
> > Thanks the authors for their responses. I think this work advances the area of medical AI. Thus I will maintain the score.

---

> > > ### Author Response · Authors · 2026-04-03
> > >
> > > Thank you for your careful reading of our rebuttal and for your valuable feedback.
> > >
> > >
> > > We will further improve the clarity of the presentation in the camera-ready version following your suggestions.

---

### Official Review · Reviewer_bqWB · 2026-03-11

**Soundness:** 3
**Presentation:** 4
**Significance:** 4
**Originality:** 4
**Overall Recommendation:** 6
**Confidence:** 4

**Summary:**

This manuscript introduces ORBIT, a Prognostic World Model aimed at the longitudinal management of ophthalmic diseases, specifically focusing on retinal conditions using optical coherence tomography (OCT). I appreciate how the authors formulate clinical disease management as a Partially Observable Markov Decision Process (POMDP), which accurately reflects the hidden pathological states over time. To address the lack of dense annotations in clinical records, the authors built a Logic-Constrained Abductive Data Engine.This uses a Propose-and-Verify paradigm to reconstruct dense pathological trajectories, resulting in the new ORBIT-LD dataset. The core framework pairs a Policy Agen with a Visual Simulator based on a diffusion model. During inference, ORBIT leverages "Counterfactual Visual Foresight" to simulate future anatomical states under various proposed treatments. This allows the model to reflect on and rectify high-risk clinical decisions before finalizing them.

**Compliance With Llm Reviewing Policy:**

Affirmed.

**Final Justification:**

My initial evaluation praised the clinical relevance of formulating longitudinal retinal disease management as a POMDP and the novelty of using visual foresight to verify treatment efficacy. However, I held reservations regarding the potential for systematic MLLM biases during dataset generation, topological drift during visual rollouts, and the fragility of the Rectify threshold.

The authors provided a brutally effective, data-driven rebuttal that resolved every single concern:
- Dataset Generation Integrity: Rather than deflecting, the authors provided a 540-sample ablation of their logic gates, proving the pipeline successfully intercepts massive clinical errors (e.g., biologically implausible temporal changes) and smartly defaults to human expert review for the rest.
- Topological Consistency: They clarified the single-step scope and provided decisive quantitative metrics (LPIPS, DINOv3) proving their Visual Simulator heavily outperforms standard image-editing baselines in preserving non-pathological retinal structures.
- Threshold Robustness: A new sensitivity analysis demonstrated that the risk landscape is sparse between 0.6 and 0.8, proving the model is not overly reliant on fine-grained hyperparameter tuning to achieve high sensitivity and specificity.

This is a methodologically rigorous paper that tackles the massive challenge of unobservable latent disease activity in OCT imaging. The authors are transparent about their limitations (compute overhead and scanner distribution reliance). I have raised my score and strongly champion this paper for acceptance.

**Key Questions For Authors:**

1. How robust is the Logic-Constrained Abductive Data Engine to hallucinated priors or systematic biases originating from the base MLLM during the "Propose" phase?
2. In the Visual Simulator, how is the structural consistency of non-pathological, rigid retinal topology maintained over longer simulated time horizons across multiple rollouts?
3. The inference pipeline includes a "Rectify" step with a heuristic adjustment if the simulated clinical risk exceeds a threshold. How sensitive is the model's overall diagnostic and treatment accuracy to the tuning of this specific hyperparameter?

**Limitations:**

While the authors provide a strong Impact Statement detailing the potential benefits and safety improvements of their framework, a dedicated discussion of ORBIT's technical limitations is somewhat lacking. The manuscript would be strengthened by a specific paragraph addressing the computational overhead of test-time visual rollouts, as well as the model's potential reliance on specific OCT scanner distributions.

**Strengths And Weaknesses:**

Soundness
- Strengths: Formulating the chronic management of retinal diseases as a POMDP is theoretically sound and well-motivated. It mirrors the clinical reality of unobservable latent disease activity. Additionally, the two-stage training curriculum (Structured SFT followed by Holistic Process-Outcome RL) for the Policy Agent is methodologically rigorous. Integrating a probabilistic transition model to simulate pharmacodynamics is a clever way to verify treatment efficacy.
- Weaknesses: My main concern is that the ORBIT-LD dataset generation relies heavily on the reasoning capabilities of an underlying LLM (Gemini 2.5-Pro). While the dual-consistency verification gates help mitigate hallucinations, there is still a lingering risk that systematic biases in the initial "Propose" phase could bypass these deterministic logical filters.

Presentation
- Strengths: The manuscript is clearly written and logically structured. The visual aids—particularly Figure 1 and Figure 3—do an excellent job of contrasting the proposed closed-loop prognostic world model against traditional open-loop black-box systems. The dataset generation pipeline is also thoroughly documented in the supplementary algorithms.

Significance
- Strengths: Shifting the medical AI paradigm from static, cross-sectional diagnosis to longitudinal, counterfactual treatment planning is a major step forward. Furthermore, releasing the ORBIT-LD dataset will provide a valuable resource for the community to explore offline reinforcement learning and world modeling in medical imaging.

Originality
- Strengths: Applying visual foresight and world models to ophthalmic OCT sequences is a highly novel combination of multimodal LLMs, diffusion models, and reinforcement learning. The idea of using generated visual futures for "closed-loop anatomical verification" to ground linguistic reasoning is a creative, impactful approach to minimizing AI hallucinations in high-stakes medical domains.

---

> ### Author Rebuttal · Authors · 2026-03-31
>
> Q1：
>
> We thank the reviewer for raising this critical question. We agree that the dual logic-gating mechanism is designed to mitigate error propagation introduced by the MLLM during the Propose phase, but it does not theoretically guarantee the complete elimination of all systematic biases. Therefore, we incorporate expert review as an additional safeguard and will clarify this limitation in the revision.
>
> To quantify this behavior, we analyze a subset of 540 samples:
>
> - (1) Decision–Biomarker Gate
>   Directly rejected 29 severe false-negative cases (clinical decision: Inject, but no biomarker detected by MLLM); flagged 123 high-risk cases (Observe with detected biomarkers) for expert review, among which 56 were confirmed as hallucinations or false detections.
>
> - (2) Temporal Plausibility Gate
>   Further filtered 35 biologically implausible “simultaneous change” sequences and 13 cases of abrupt biomarker burden changes.
>
> | Stage | Trigger Condition | Action | Count |
> |------|------------------|--------|------|
> | Propose | MLLM generates candidate report/biomarker | Proceed to validation | 540 |
> | Decision–Biomarker Gate | Inject but no biomarker | Reject & regenerate | 29 |
> | Decision–Biomarker Gate | Observe but biomarker present | Flag for expert review | 123 |
> | Expert Review | Within high-risk samples | Confirm hallucination / error | 56 |
> | Temporal Gate | Implausible simultaneous change | Filter & review/regenerate | 35 pairs |
> | Temporal Gate | Abrupt biomarker burden shift | Filter & review/regenerate | 13 pairs |
>
> These results demonstrate that the dual logic gates effectively intercept two major categories of high-risk errors in the data flow: (i) decision–biomarker inconsistency, and (ii) biologically implausible temporal evolution. From a bias perspective, this indicates that systematic errors explicitly manifested through these patterns can be significantly reduced. However, we cannot guarantee the complete elimination of more subtle biases that do not violate current logical constraints. Therefore, we retain expert review as an additional safety mechanism.
>
> ---
> Q2:
>
> The visual simulator generates future-stage images conditioned on the current-stage source image and treatment actions, thereby anchoring anatomical structure to the source image. Training supervision is derived from real longitudinal image pairs, enabling the model to capture pathological changes while preserving non-pathological retinal topology.
>
> This work focuses on short- to mid-term single-step (next-visit) prognosis rather than long-horizon multi-step rollouts. Within this scope, conditioning on the source image effectively suppresses non-pathological topological drift.
>
> We therefore added quantitative evaluation on a test set of real longitudinal image pairs.
> | Metric | Qwen-Image-Edit | Visual Simulator |
> |---|---|---|
> | SSIM ↑ | 0.1445 | 0.3556 |
> | PSNR ↑ | 9.82 | 24.32 |
> | LPIPS ↓ | 0.8400 | 0.4018 |
> | DINOv3-FID ↓ | 37.62 | 2.761 |
> | DINOv3-KID (×1000) ↓ | 72.50 | 3.27 |
> | Biomarker agreement ↑ | - | 0.8121 |
>
> ---
>
> Q3:
>
> We thank the reviewer for this important question. ORBIT assigns each sample a continuous risk score \(R_{\text{risk}} \in [0,1]\) via rollout simulation, and we set the Rectify threshold to **0.7** in all experiments.
>
> To assess sensitivity to this choice, we further analyzed the test-set risk distribution and evaluated performance under different \$\tau_{\text{risk}}$. We found that the risk landscape is well structured: high-risk and low-risk cases are clearly separated, while **0.6–0.8** forms a relatively sparse transition region. Consequently, varying $\tau_{\text{risk}}$ within this range changes the set of rectified samples only marginally, leading to stable overall performance.
>
> | $\tau_{\text{risk}}$ | Acc | Sens | Spec |
> |---|---:|---:|---:|
> | 0.4 | 83.70 | 86.46 | 80.56 |
> | 0.6 | 88.33 | 82.99 | 94.44 |
> | 0.7 | 88.52 | 82.29 | 95.63 |
> | 0.8 | 88.15 | 80.90 | 96.43 |
> | 0.9 | 78.15 | 59.72 | 99.21 |
>
> This suggests that $\tau_{\text{risk}}$ is not a highly sensitive hyperparameter. ORBIT remains stable throughout the **0.6–0.8** range, while more extreme thresholds show the expected trade-off: lower thresholds are more conservative, improving sensitivity at the cost of specificity, whereas higher thresholds are more likely to miss progression. Overall, **0.7 is a robust default choice** that achieves a good balance between safety and intervention accuracy without requiring fine-grained tuning.
>
> ---
>
> limitation:
> We thank the reviewer for this suggestion. In the revision, we will add a dedicated limitations paragraph discussing:
> (i) the additional computational overhead introduced by test-time visual simulation and reflection;
> (ii) the potential dependence on scanner-specific OCT image distributions, despite encouraging OOD performance on OLIVES.

---

> > ### Author Rebuttal · Reviewer_bqWB · 2026-04-02
> >
> > Authors often try to hand-wave concerns regarding LLM hallucinations and structural drift, but this rebuttal was exceptionally rigorous and data-driven. You provided exactly the quantitative evidence required to back up your claims.
> >
> > - Regarding the Logic-Constrained Abductive Data Engine (Q1): The breakdown of the 540 samples is highly illuminating. Admitting that the logic gates cannot eliminate all systematic biases and retaining the expert review as a final safeguard is the right clinical approach. The fact that the temporal gate explicitly caught 35 biologically implausible sequences proves the absolute necessity of your pipeline over naive MLLM generation.
> >
> > - Regarding structural consistency (Q2): Clarifying the scope as single-step (next-visit) prognosis is a vital distinction. The new quantitative metrics (especially the LPIPS and DINOv3 comparisons against Qwen-Image-Edit) definitively prove that conditioning on the source image suppresses non-pathological topological drift.
> >
> > - Regarding the Rectify threshold (Q3): The sensitivity analysis is precisely what I was looking for. Demonstrating that the risk landscape is sparse between 0.6 and 0.8, and thus not overly sensitive to hyperparameter tuning, gives me strong confidence in the model's robustness for real-world inference.
> >
> > You have fully resolved my concerns. The integration of a prognostic world model for longitudinal retinal OCT is a highly significant contribution. I expect the new quantitative tables (Q2 and Q3) and the explicit limitations paragraph regarding compute overhead and scanner-specific distributions to be fully integrated into the camera-ready manuscript.

---

> > > ### Author Response · Authors · 2026-04-03
> > >
> > > Thank you very much for your valuable feedback and for raising your score!
> > >
> > >
> > > In the camera-ready version, we will incorporate the new results (Q2, Q3), expand the discussion on computational overhead and scanner dependency, and further clarify the role of expert review in mitigating residual bias. We sincerely appreciate your recognition and insightful feedback.

---

### Decision · Program_Chairs · 2026-04-30

**Decision:**

Accept (regular)

**Comment:**

The paper studies longitudinal management of blinding fundus disease. It frames the task as a POMDP and proposes ORBIT, a prognostic world model that couples a policy agent with a diffusion-based visual simulator. It also introduces ORBIT-LD, a longitudinal decision dataset built with a propose-and-verify pipeline to recover dense biomarker and evolution labels from sparse clinical records. The main idea is to check treatment choices against simulated future anatomy, not only against text reasoning.

The paper is clearly structured and easy to follow. Its POMDP framing is sound and clinically well motivated, the dataset construction is substantial and gives the work a real benchmark, and the closed-loop simulator provides a concrete mechanism for comparing candidate actions against future anatomy rather than relying only on text-based reasoning. The rebuttal also added useful quantitative evidence for simulator fidelity and threshold behavior.

The authors engaged well and answered the technical questions with concrete additions. They added quantitative checks for the data gates, the simulator, and the threshold sweep, and they were explicit about what remains future work. The exchange moved the reviews in a material way.

The paper will be read by researchers on medical decision support, world models, and offline RL, and it will likely be cited for ORBIT-LD and the closed-loop anatomical verification framing. The real reason to accept is that it turns sparse longitudinal OCT records into both a benchmark and a decision system with measurable gains in perception, evolution, and treatment planning. The open items are real, but they are limits of scope rather than failures of the core idea. The paper contributes a new benchmark and a concrete method for counterfactual decision making under partial observability.